# Data Verification in the Agent, Combining Blockchain and Quantum Keys by Means of Multiple-Valued Logic

## Alexey Yu. Bykovsky * and Nikolay A. Vasiliev

P.N. Lebedev Physical Institute RAS, Leninsky pr. 53, 119991 Moscow, Russia
* Correspondence: bykovskiyay@lebedev.ru

**Abstract:** Network control of autonomous robotic devices involves a vast number of secured data coding, verification, and identification procedures to provide reliable work of distant agents. Blockchain scheme provides here the model of the extended linked list for the verification of critical data, approved by quasi-random hash values assigned by external network nodes. And quantum lines are the source of high-quality quasi-random keys used as hash values. Discrete multiple-valued logic in such procedures is a simple and flexible tool to form the logic linked list, combining critical internal parameters of agents with data taken from external nodes. Such combination enlarges the set of possible schemes for data protection from illegal modifications and for data restoration.

**Keywords:** data verification; agent; multiple-valued logic; blockchain; linked list; quantum key





## 1. Introduction

Modern concepts of communication networks include such collective robotic components as autonomous vehicles (or Internet of Vehicles, IoV) [1,2], logistics [3], industrial machinery [4], and Internet of Things (IoT) [5,6]. As autonomous agents are based on traditional microprocessors and embedded platforms [7], then attacks of eavesdroppers are quite possible and efficient distant monitoring plays critical role. Discussions of possible methods for distant data verification has included special network protocols for mobile agents [8–10], quantum cryptography [11,12], schemes of distributed computing [13], blockchain (BC) methods [14–16], position-based cryptography [17], and non-binary k-valued discrete logic [18]. Interest to blockchain technologies for network agents can be attributed to the problem of low availability of trusted network nodes for mass robotics, as trusted systems for communication networks discussed e.g., in [19] need special equipment, personnel, and intensive monitoring procedures. At the same time, the integration of fiber–optics and wireless quantum key distribution (QKD) lines [12,20] in communication networks provides new possibilities to raise security, although QKD is a complicated and expensive tool, which has its specific vulnerabilities and can be suppressed by intensive optical noise. QKD schemes and quantum random number generators (QRNG) [21] are the high quality sources of one-time quasi-random keys, which provide the base for various data protection schemes. That is why they are also expected to raise the security level of autonomous robotic systems based on artificial intellect (AI) methods [22], which are strongly associated with the concept of collectives of intellectual agents or multi-agent systems (MAS) [23]. In Figure 1 a local network of hardware agents 0, 1, . . . is a part of nodes of the communication network, and can include mobile and static agents. As autonomous agents should be able to work without permanent personnel control due to the imitation of human abilities by means of AI algorithms [19], various verification procedures are quite actual for them, including schemes based on BC [24–26]. However, data protection of internal subsystems in autonomous agents is a separate problem, as an agent, see Figure 1 to the left, may include subsystems with different throughput, software and hardware complexity [27]. As a result, hardware platform for autonomous agents can be very complicated [27,28], and model debugging and data verification is the difficult problem.

**Agent`s internal components**:
power supply,
decision-making module,
communication module,
cryptographical module
sensors,
computer vision system,
positioning devices,
actuators,
data buses, etc.

**Figure 1.** Communication network includes usual stationary nodes and autonomous mobile robotic agents 0, 1, . . . , which are the participants of a multi-agent system (MAS) and can follow some given route like agent 1. Nodes called checkpoints are to approve identity of the mobile agent and to give access to a requested service. Such nodes optionally may be participants of the trusted local network.

For the position–based cryptography task [17] and agents following the given route, verification procedures [29] should combine authentication, identification and secret coding. It is necessary to check identity and access rights of the mobile agent by verification of parols, pin-codes, licenses, digital images, and prehistory of carried out agent's tasks, what involves either the design of trusted check-points [29] or needs the estimation of trust models [30,31]. Besides this, schemes based on BC principles [32–34] were already designed for the collective verification of data in energy power networks, as static network agents of dishonest users can declare false data.

## 1.1. Modern Platforms for Robotics Positioning and Communications

Localization or positioning was considered in reviews [35,36] as a process to obtain tracked objects information concerning multiple reference points within a predetermined area, i.e., it is a procedure to identify the position of mobile/fixed devices, including smartphones, drones, watch, beacons, and vehicles using some fixed nodes and mobile computing devices. Global navigation satellite system (GNSS) in such tasks can use signals produced by global positioning system (GPS), global navigation satellite system (GLONASS), Galileo and Beidou. As the GPS device loses substantial power in an indoor setting due to signal attenuation, such systems cannot be used for indoor localization of devices. Alternative possibilities refer to ZigBee, Bluetooth, Radio Frequency Identification (RFID), cellular networks (including LTE and 5G), ultrawideband (UWB), frequency modulation (FM), inertial sensors and Wi-Fi. Some hybrid approaches are also possible. Most common techniques for localization include time-of-flight (TOF) measurements and received signal strength indicator (RSSI) signal measurements, utilizing the distance between known fixed stations and the target device, or fingerprint-based location estimation.

Vehicular ad-hoc network (VANET) [37] was based on the link-layer communication (IEEE 802.11p) and has included the data exchange between the high-speed vehicles in the licensed band of 5.9 GHz (5.850–5.925 GHz). VANET differs from other ad-hoc networks by high mobility, dynamic topology, frequent data exchange, unbounded network size, unlimited battery power, and predictable movement (which happens only on the road). It may use two types of nodes: (1) mobile nodes attached to onboard units, and (2) static nodes like traffic lamp posts, signboards, and roadside units. This network of mobile agents has provided connections vehicle-to-vehicle, vehicle-to-infrastructure, and infrastructure-to-

infrastructure. Used performance metrics were hop length, minimum energy, link lifetime, route breakage, and bandwidth.

Very actively investigated field of autonomous robotics refers to flying unmanned aerial systems (UAVs). The authors of the review [38] have analyzed flying ad-hoc networks (FANETs), which can be deployed either individually or may be incorporated into traditional wireless local area networks (WLANs). Its main application fields include search and rescue, mailing and delivery, traffic monitoring, precision agriculture, and surveillance applications. Unmanned service of FANETs is actual in case of natural disasters, hazardous gas intrusions, wildfires, avalanches, and search of missing persons. As routing is the most challenging job in FANETs due to such attributes of UAVs, as high mobility, 3D movement, and rapid topology changes, then a predictive method should be used for path planning and navigation in order to prevent possible collisions and to ensure the safety of the FANET. However, the data aggregated by a small UAV can be too large to be processed and stored onboard [39]. Small UAVs in FANETs also suffer from security vulnerabilities, as their limited storage and computing capabilities do not allow to perform computational-intensive tasks locally [40,41]. An intruder intending to attack the network UAV [38] can transmit massive reservation requests, eavesdrop instructions, and modify the information. UAVs connected to Wi-Fi are considered as less secured in comparison with cellular networks, due to unreliable links and poor security methods. False transmitter can be attached to a UAV and may send fake instructions, in addition to this UAVs can become a luring target for physical attacks [39,41]. In such instances, an attacker can dissemble the captured UAV to get access to internal data via interfaces and USB ports.

GPS spoofing, see e.g., [38] is another major security threat for small UAVs. An adversary can transmit fake GPS signals to an intended UAV with enlarged power than the actual GPS signals. Thus, localization system must verify actual positions of neighboring UAVs and associated distances in order to avoid the GPS spoofing attacks.

The detection and identification of vulnerabilities for UAVs refers to popular short-range wireless networking technologies like Wi-Fi (IEEE 802.11), ZigBee (IEEE 802.15.4), Bluetooth (IEEE 802.15.1), LoRaWAN, and Sigfox [38,42], differing by the range and the throughput for licensed or unlicensed spectrum types. Wi-Fi provides a set of specifications for radio bands of 2.4, 3.6, 5, and 60 GHz. IEEE 802.11a/b/g/n/ac is the first choice to provide the transmission of medium size video and images for distances of approximately 100 m, but unlicensed versions can provide up to few hundred meters. A multi-hop networking scheme may expand the transmission range to kilometers. An alternative to Wi-Fi is the use of low-cost and low-power methods like Bluetooth and ZigBee. Bluetooth (IEE 802.15.1) is a low cost and low power variant, which operates in an unlicensed band of 2.4 GHz with a contact range of 10 to 100 m and uses a distributed frequency-hopping transmission spectrum.

Licensed 5G and 6G generation technologies are expected [38] to offer improved data rates and coverages in linking of FANETs, to provide high device mobility and integration of a massive number of UAVs in an ultra-reliable way, to serve multi-access edge computing, and to incorporate cloud computing. Low-power wide area networks (LPWAN) can be another good option for UAVs which consumes less energy and offers a wide range of connectivity. LPWAN allows transmitting data for a longer duration of time and without much loss of energy resources. For IoT users, LoRaWAN [43] has been designed as a technology for the management of low energy consumption transmissions, using a novel network paradigm for bidirectional connectivity, localization, and mobility management. It provides a new framework for LPWAN execution, providing long-range communications in the band 868/900 MHz with data rates ranging from 0.3 kbps to 50 kbps and network coverage from 5 to 15 km. Sigfox, similar to LoRaWAN, is a low-speed, but low-power and long-range solution for UAVs, it supports open-sight up to 30 km of range in the same band as LoRaWAN.

Another modern trend for enhancement of data privacy and integrity in UAV communication networks is the aerial blockchain, especially supported by 5G and 6G [38,44].

Blockchain-based software for UAV is expected to provide flexibility, dynamics, and on-the-fly decision capabilities. UAVs can be integrated potentially into the Internet network, providing access to cloud computing and web technologies for the realization of smart IoT systems.

### 1.2. Blockchain-Based Verification Schemes for Network Agents

Principally, methods of data verification for agents and MAS can include various traditional technologies [12], including parity bits checking, cryptography algorithms and biometrics, as well as exhausted methods of model checking [45], temporal logic and Kripke diagrams [46,47]. Unfortunately, known BC schemes [14,16,32] designed for crypto currencies mining scarcely can be directly used for data protection of robotic agents and their internal subsystems, mainly due to energy expenses, high cost, limited throughput of microprocessors, and incompatibility of specialized technical tasks of robots with basic Proof-of-Work and less expensive Proof-of-Sake schemes [48,49]. Respectively, the design of secured and reliable agent's devices for mass robotics, IoT, and IoV stimulates the search of more simple and flexible verification schemes for distant monitoring of static and mobile agent's behavior. It is substantial, that BC-based schemes for trust parameters estimation of stationary agents in power grid networks [32–34] has already demonstrated the effectiveness of collective detection of dishonest user's nodes by means of polling of neighboring nodes and evaluation of the trust parameters by collaborating network nodes.

### 1.3. Multiple-Valued Logic for Data Verification Based on Blockchain Scheme

Multiple-valued logic (MVL) version of the linked list [19] has used partially the BC scheme [14–16,32–34,48,49] and was aimed at data verification in case of faults. Such method bases on original property of $k$-valued Allen-Givone algebra (AGA) [50] to aggregate correctly arbitrary variables into logic functions, to form logic linked lists of entries, and to obtain protective logic expressions, preventing from illegal modification of logic product terms. It is substantial, that MVL-based linked list (MVLL) [19] can also model quantum verification protocol of position-based cryptography [51], thus demonstrating the possibility to combine quantum and classic verification procedures. Further research of AGA model [19,29] was also motivated by some earlier papers like [52,53] and e.g., by MVL scheme [54] based on lattice models and the theory of sets.

MVLL [19] is the model for distributed or collective data storage of entries $e = \{e_1, \ldots, e_p\}$, containing sets of $p$ critical parameters of the mobile or the stationary agent, verified by collaborating agents and network nodes, participating in the collective protocol. Here the difference between collaborating agents and nodes mainly refers to levels of trust assigned to interacting devices. e.g., for route verification task trusted check points always should be assigned the highest trust and priority level, as other collaborating network nodes are motivated to take part in the joint protocol by mutual service or fee, and should be appreciated by lower trust values. But before the detailed design of trust estimation schemes for mobile robots one should work out adequate verification procedures involving all possible resources of agents and network nodes.

### 1.4. Internal and External Data Verification Procedures of the Network Agent

Distant control of autonomous robotic agents involves at least two types of tasks greatly differing by the time response. Such basic AI tasks as e.g., logistics, building work, following the route, interaction with check-points, takes long time intervals (days, weeks, months etc.) and the time moment for the end of task can't always be predicted precisely. Also verification procedures for digital licenses and documents may be requested optionally by other participants of the network. In contrast to this, the second type of control processes refers to internal systems of the agent, which typically use rigid short time work cycles (μs, ms, s) [55] for control of numerous sensors and actuators. Thus, complex data verification procedure of the autonomous agent should include both types of time processes. In order to make verification data unpredictable for possible eavesdroppers, verified parameters

should be masked or even secretly coded by one-time quasi-random keys, traditionally used in the most reliable one-time cipher pad method of secret coding [56], which is the final aim of any QKD line realization [12]. For internal systems such quasi-random codes also can be done one-time and actual only for short time intervals, i.e., they should be renewed for a selected or every new work cycle. Then quasi-random keys are preferably to be generated by either a QKD line [20], or by a quantum random number generator (QRNG) [21] and random oracle [57] for simplified and less protected variants. Used for verification hash values from external and internal components of the agent are to be mixed with technical parameters written in the distributed linked list. But in contrast to blockchain schemes in cryptocurrency mining [25,26,48,49] and software bots for stock trading [58], autonomous hardware robots can provide very limited resources for additional verification as they are initially intended to carry out highly specialized work. Thus, the possibility of data verification inside the agent strongly depends on its internal structure of subsystems, whose most typical examples are listed in Figure 1 to the left. Here the simple way to unite various internal and external verification schemes in one agent seems to be based on the earlier proposed MVLL model of the distributed ledger or the linked list [19], which is to be additionally approved by some internal and external technical parameters besides externally assigned hash values.

### 1.5. Possible Sources of Quasi-Random Keys for Hashing Procedures

As any computerized device the distant autonomous agent can be damaged by a physical impact on its embedded computer, or can be infected by a computer virus [12]; possible attacks [59,60] can be also aimed at internal data buses, memory and service devices. These hardships makes actual involvement of QKD lines, as the interest to quantum keys for wireless network robotics bases on the principles of modern cryptography, according to which the well-known Vernam code or close to it one-time cipher pad method [56] with random one-time keys is the most reliable cryptography scheme. Due to the non-cloning theorem [61] QKD lines are in fact the way to exclude long-range distant storage of secret keys by means of their quick enough generation for every new communication session. But as the throughput of quantum lines is not enough for massive stream secret coding, and its length is limited by ~100 km [12], this method has now mainly nice applications, its drawbacks are the high cost and the possibility of suppression by intensive noise, what is especially actual for wireless robotics communications. Also QKD line itself can be vulnerable to some specific types of quantum and classical attacks [12], and the access of non-loyal personnel to agent's equipment can cause data leakages. Besides this, attempts to design purely quantum verification schemes for position-based cryptography in mobile systems [17,62] has given only partial improvement and unconditional security was not still obtained, as well as quantum bit commitment schemes still were not realized. Also it is the too expensive and complicated tool to be used in internal data buses in the agent. However, the research of QKD lines [20] has drawn conjugated investigations of quantum random number generators (QRNG) [21] and the position-based cryptography (PBC) scheme [57], involving random oracle model [17] based on memory device and MVL function learned by a QRNG. Such method was adapted for the verification of visits of mobile agents to check-points, conjugated with the quantum protocol by D. Unruh [51] using entangled photon pairs. The principal possibility to apply random oracle as a more simple alternative source of quasi-random keys for MVLL is a way to design new verification procedures and to avoid partially the use of trusted checkpoints. However, before the design of future schemes to estimate the trust level of mobile agents and static check-points, one should work out effective non-quantum methods to keep reliably critical data in distant agents and consequently to verify data written in them, as well as to check prehistory of agent's work activity and passed routes.

### 1.6. Heterogeneous Logic Models in Modern Controllers

The motive to apply heterogeneous logic models in the agent [18,19] and to combine Boolean and MVL logic in it is determined by the fact, that verification procedures in a robotic agent refers not only to PC's microprocessors, but also to controller devices based on the closed loop control and equipped with own microprocessors [55] with lower throughput. Traditional types of such devices are mostly represented by proportional-integral-differential (PID) [63] and fuzzy logic controllers [64–66]. However, modern investigations reveal the stable trend to further integration of PID and fuzzy logic in controllers, see e.g., [64]; respectively, this paper has demonstrated the actuality of heterogeneous logic models not only for agents level [18], but also for the level of internal controllers. The so-called fuzzy-fractional order-proportional-integral-differential controller (FFOPID) in [64] has provided improved non-linear characteristics modelled by non-linear polynomials. Fuzzy logic in this FFOPID scheme has provided fast approximate computing and simple learning of the system, but has needed correct emulation of specific operators of the fuzzy logic.

### 1.7. The Goal of the Paper

The final goal of the proposed further extended version of the MVLL [19] is to combine more complicated and intellectual robotic data procession procedures with BC-based schemes, secured from illegal modifications and capable for self-checks and self-restoration in case of fails and faults. The scheme disclosed in this paper is the step to approve critical task parameters and control signals of internal subsystems of the agent by external network nodes. Such distributed storage of verification data is necessary in cases, when data taken from different sources contradict each other and need detailed verification. Then the sequential matching of data copies extracted from external and internal backups can help to restore reliable work and avoid blocking of the robot.

The aim of the proposed paper is to design the new data verification scheme for mobile agents, complementing earlier proposed version of the MVLL by the documented data exchange between external nodes and internal data storages. The expected advantage here is the diversity of verification methods in case of faults and errors.

## 2. Methods: MVL Linked List as the Data Protection Model

MVLL method [19] bases on discrete-valued logic and partially uses the idea of mixing data blocks in BC schemes by combining quasi-random hash values with real values of variables in the last and in one of previously formed entries.

### 2.1. Logic Functions of the Multiple-Valued Allen-Givone Algebra

Detailed description of basic calculations within discrete $k$-valued Allen-Givone algebra (AGA) [50] was given in, e.g., [18,19]. The choice of this version of logic calculus was determined by the simplicity of its basic operators and the flexibility of design of multiparametrical functions. In contrast to binary Boolean logic, AGA function $y = f(x_1, \ldots, x_n)$ can be given by $n$ input variables $x_1, \ldots, x_n$ and one output variable $y$, which may be assigned $k$ discrete truth levels, i.e., $x_1, x_2, \ldots, x_n, y \ L = \{0, 1, \ldots, k - 1\}$. The complete set of its logic operators [50]

$$< 0, 1, \ldots, k - 1, \ X(a, b), \ \star, + > \qquad (1)$$

guarantees the possibility to represent arbitrary function $y = f(x_1, \ldots, x_n)$ as some combination of logic constants $0, 1, \ldots, k - 1$, binary operators $\mathrm{Min}(x_i, x_j)$ marked by ($\star$) and $\mathrm{Max}(x_i, x_j)$ marked by (+); also unary operator $X(a, b)$ is being used, which is called Literal. Operators Min and Max, respectively, choose either the minimal value in the pair $x_i$ and $x_j$, or the maximal one. Literal is defined as Exp. (2)

$$X(a, b) = \begin{cases} 0, & if \ b < x < a \\ k - 1, & if \ a \leq x \leq b \end{cases} \qquad (2)$$

where for any $X(a, b)$ always $b \geq a$, and $a, b \in L = \{0, 1, \ldots, k-1\}$.

For verification procedures the advantage of AGA [18,19] is namely the guaranteed possibility to obtain correct logic expression for arbitrarily given data, where the only possible algorithm for calculation of MVL functions excludes alternative illegal procedures for eavesdroppers. The drawback of such functions is some unpredictability of calculation time for unknown sets of data, as the minimization for MVL is very wasteful [50].

MVL truth table [50], see Table 1, partially resembles the Boolean ones, but contains much more rows whose number attains $k^n - 1$, where $k$—is the number of discrete logic levels, and $n$ is the number of input variables. If one has composed MVL truth table, then every its row has equivalent logic expression written to the right of Table 1. The column for output variable $F(x_1, \ldots, x_n)$ should be arbitrarily filled in by logic constants from the set $C = \{0, 1, \ldots, k-1\}$. Respectively, if constant $F$ is 0, then the product term of this row is also equal to 0. Note that Literal operators to the right to Table 1 includes equal values $a$ and $b$.

**Table 1.** Truth table of a AGA function. Equivalent product term for every row with nonzero output value ) is given to the right.

| $N_{row}$ | Input Variables | | | | | | Output Variable | |
|---|---|---|---|---|---|---|---|---|
| | $x_1$ | $x_2$ | $\ldots$ | $x_{n-1}$ | $x_n$ | $F(x_1 \ldots x_n)$ | | **Equivalent Product Terms:** |
| 0 | 0 | 0 | $\ldots$ | 0 | 0 | $F(0, 0, \ldots, 0)$ | | $F(0, 0, \ldots, 0) \star X_1(0,0) \star X_2(0,0) \star \ldots \star X_n(0,0)$ |
| 1 | 1 | 0 | $\ldots$ | 0 | 0 | $F(1, 0, \ldots, 0)$ | $\rightarrow$ | $F(1, 0, \ldots, 0) \star X_1(1,1) \star X_2(0,0) \star \ldots \star X_n(0,0)$ |
| $\ldots$ | $\ldots$ | $\ldots$ | $\ldots$ | $\ldots$ | $\ldots$ | $\ldots$ | | $\ldots$ |
| $k^n - 1$ | $k-1$ | $k-1$ | $\ldots$ | $k-1$ | $k-1$ | $F(k-1, \ldots, k-1)$ | | $F(k-1, k-1, \ldots, k-1) \star X_1(k-1, k-1) \star$ $X_2(k-1, k-1) \star \ldots \star X_n(k-1, k-1)$ |

Resulting Exp. (3) can be written as:

$$F(x_1, \ldots, x_n) = F(0, 0, \ldots, 0) \star X_1(0,0) \star \ldots \star X_n(0,0) +$$
$$+ F(0, 0, \ldots, 1) \star X_1(1,1) \star \ldots \star X_n(0,0) +$$
$$+ F(k-1, k-1, \ldots, k-1) \star X_1(k-1, k-1) \star \ldots \star X_{jn}(k-1, k-1). \quad (3)$$

MVL minimization procedure by means of consensus method [40] involves transformation of parameters $(a, b)$ in Literals and can shorten the number of product terms, but it is a wasteful enough procedure, which bases on the subsuming of product terms and the method of "don't care states" [50]. One can find its detailes in [18].

If one use equivalent matrix representation of MVL function [29], which was formed by data taken from the truth table (Table 1), then matrix C in exp. (4) will have only one-column, and exp. (3) can be written as arrays

$$A = \begin{pmatrix} a_{11} & \ldots & a_{1n} \\ \ldots & \ldots & \ldots \\ a_{k^n-1, 1} & \ldots & a_{k^n-1, n} \end{pmatrix}, B = \begin{pmatrix} b_{11} & \ldots & b_{1n} \\ \ldots & \ldots & \ldots \\ b_{k^n-1, 1} & \ldots & b_{k^n-1, n} \end{pmatrix}, C = \begin{pmatrix} F(0, \ldots, 0) \\ \ldots \\ F(k-1, \ldots, k-1) \end{pmatrix}, \quad (4)$$

where $b_{ij} \geq a_{ij}$, $n$—the number of input variables, $k$—the number of truth levels. Matrixes $A_u$ and $B_u$ in Exp. (4) define parameters $a$ and $b$ in Literals $X_j(a, b)$. Due to very large possible number of rows in the MVL truth table [50], the real system scarcely can use all possible $k^n$ rows.

As MVL minimization by consensus method [50] for composed MVL function can shorten the number of product terms and change parameters in matrixes A,B,C, it can hide illegal modifications done in memory. In any way, all possible transformations of AGA expressions are based on the subsuming property of MVL product terms [50].

**Definition 1,** cited from [50]. *Product term $r_1 \star X_1(a_1, b_1) \star \ldots \star X_n(a_n, b_n)$ subsumes another product term $r_2 \star X_1(c_1, d_1) \star \ldots \star X_n(c_n, d_n)$, if and only if conditions (1) and (2) are true:*

(1)  $r_1 \leq r_2$,

(2)  $c_i \leq a_i \leq b_i \leq d_i \text{ for all} X_i, i = 1, \dots, n$.

For examplelogic expression taken for k = 256 truth levels

$\cancel{57 * X_1(24, 24) * X_2(127, 1275) * X_3(317, 331)} +$

$+80 * X_1(24, 24) * X_2(1275, 1275) * X_3(317, 331) =$

$= 80 * X_1(2, 24) * X_2(1275, 1290) * X_3(317, 331)$  contains three product terms with constants 57, 80. As according to condition (1) in Definition 1 relation $57 \leq 80$ is true, and according to condition (2) $c_i = a_i \leq b_i = d_i$  for all $X_i$, $i = 1, 2, 3$, respectively the first product term (shown above) will subsume the second one and can be crossed out. That is why data protection for AGA model need monitoring of the number of product terms and values of parameters in matrixes A, B, C.

### 2.2. Possible Attacks Aimed at MVL Product Terms and the Role of Subsuming

As cheaters potentially can modify AGA formalisms written in the memory of the agent, then all possible illegal modifications of MVL function are grouped in Table 2.

**Table 2.** Possible attacks aimed at MVL functions. (Abbrev. PT—tags product term).

| N | Attack Type | Consequences, Scheme of Protection |
|---|---|---|
| 1. | Adding of one new PT with fake logic constant $C*$ and previously used real parameters $a, b$. | (1)  If $C^* \geq C$ (*e.g.*, $38 > 12$), then $$PT_{real} + PT_{fake} = \cancel{12 * X_1(2, 4) * X_2(75, 175) * X_3(31, 34)} + +38 * X_1(2, 4) * X_2(75, 175) * X_3(31, 34).$$ Result: Subsuming of the real PT by the fake one and incorrect logic values for real data of variables $x_1, x_2, x_3$. Protection: Monitoring of the set of logic constants C for all PTs. (2)  If $C^* < C$ (*e.g.*, $11 < 12$), then $$PT_{real} + PT_{fake} = 12 * X_1(2, 4) * X_2(75, 175) * X_3(31, 34) +$$ $$\cancel{+11 * X_1(2, 4) * X_2(75, 175) * X_3(31, 34).}$$ Result: Guaranteed subsuming of the fake PT by the real one. No danger! |
| 2. | Adding of one new product term (PT) with real constant $C$ and fake parameters $a^*, b^*$. | $$PT_{real} + PT_{fake} = \cancel{12 * X_1(2, 4) * X_2(75, 175) * X_3(31, 34)} + +12 * X_1(2, 6) * X_2(75, 177) * X_3(31, 36).$$ Result: Subsuming of the real PT by the fake one, logic value 12 will be incorrectly assigned to the widened set of values for variables $x_1, x_2, x_3$, but not only to real ones. Protection: 1.  Method of blocking logic terms [19] for $a^*, b^*$ beyond the real band. 2.  Monitoring of a set of of real parameters $a, b$. |
| 3. | Adding of one new product term with fake logic constant $C$ and fake values $a^*, b^*$. | Result: Simultaneous impact of NNs 1 and 2 given above. Protection: 1.  Method of blocking logic terms for $a^*, b^*$ beyond the real band. 2.  Monitoring of sets of real logic constants C and parameters a,b. |
| 4. | Deleting of the PT, i.e., replacement of the logic constant C by $C^* = 0$ in PT. | Result: Deleting of the whole PT will cause the change in the overall number $N_{pt}$ of PTs in the MVL function. Protection: Monitoring of numbers: $m$ of product terms, real logic constants C, and parameters $a, b$. |

**Table 2.** *Cont.*

| N | Attack Type | Consequences, Scheme of Protection |
|---|---|---|
| 5. | Replacement of the logic constant $C$ by fake $C^* \neq 0$ in some PT, parameters $a, b$ are real. | Result: Values of fake $C^*$ are assigned to correct values of a,b: $\quad PT_{fake} = C^* * X_1(2,4) * X_2(75,77) * X_3(31, 34)$. Protection: Monitoring of sets of real constants C and parameters $a$, $b$. |
| 6. | Replacement of real $a$, $b$ by fake ones $a^*$, $b^*$ in Literals $X_i(a,b)$ of one PT. | Result: Incorrect values of C are assigned to fake values. Possible protection: Method of blocking logic terms for $a^*$, $b^*$ beyond the real band. |

Product terms, subsumed after possible attacks are shown crossed out in Table 2. The analysis of given schemes of attacks aimed at MVL formal expressions shows the necessity to combine at least two basic schemes for data protection. Three types of attacks (NNs 2,3,6 in the Table 2) can be protected by the earlier proposed method of blocking product terms [19], describing all possible intermediate rows of the truth table located between two given rows. However, other types of attacks 1,4,5 need methods to monitor integrity of parameters of MVL functions. But here MVL version of BC-based schemes gives the method to form the protected from modifications linked list with confirmation by distributed external and internal parameters. Advantage of AGA functions calculations is that they are based on logic primitives and always use the only possible algorithm, thus they has no specific vulnerabilities for traditional attacks [12,18,19].

*2.3. MVL Scheme of the Linked List*

Principally, initial MVLL scheme [19] includes two protective tools:

(1) duplicating notation of the same message in the last and in the previous entries like in BC schemes [25,26], and (2) the approval of data by quasi–random values of the hash function, assigned by the set of external distant nodes. Any attempt to modify data in the MVLL will need to modify at least two rows in the truth table of the linked list, approved before by two sets of externally assigned hash values. That is why the first way to use MVLL is to detect attacks 1,4,5 in Table 2 by comparison of entries, written in internal and different external parts of the linked list. The second way is to use blocking terms [19] preventing from 2,3,6, types of attacks.

In order to write the new entry $e$ to MVLL [19], the mobile agent (called the prover in route verification tasks [41]) declares the new entry to $Q$ collaborating nodes (or verifiers) by sequential mailing according to the list of participants of the protocol. First of them assigns the time stamp $t$, and each of verifiers should assign its quasi-random hash value $h$ to the received entry $e$. These hash values are to be preliminary generated by QKD lines and accumulated in nodes in order to provide maximal unpredictability of data. One of assigned hash values should be used as the output value of the MVLL function, confirming every entry in the MVLL. After the acquisition of external hash values, mobile agent sends the complete set of data $m$, $t$, $e_m = \{e_{1,m}, \ldots, e_{n,m}\}$, and $h = \{h_1, \ldots, h_Q\}$ to external participants (i.e., "witnesses") for their backup storage copies, formed individually according to known common rules. The absence of the verifier's reply due to faults and switched–off mode is to be fixed by zero.

In general case one can form the linked list function $F_{ll}$ [19] by product terms, containing pairs of entries obtained at different time moments $t_m$ and $t_{m-s}$, where $m$ is the number of the last entry; $s$ is the shift of the number for the previous entry. Function $F_{ll}$ responds to Definition 1 and Exp. (5).

**Definition 2.** cited from [19]. *Logic ledger function or a linked list of logic entries is given within AGA as a hash function*

$$
h^{(out)} =
F_{ll}\left(m, s, t, e_{1,m}, \ldots, e_{p,m}, e_{1,m-s}, \ldots, e_{p,m-s}, h_{1,m}, \ldots, h_{Q,m}, h_{1,m-s}, \ldots, h_{Q,m-s}\right),
\tag{5}
$$

or in the compact notation $h^{(m,s)} = F_{ll}(m, s, t, \boldsymbol{e_m}, \boldsymbol{e_{m-s}}, \boldsymbol{h_m}, \boldsymbol{h_{m-s}})$.

In Exp. (5) $m$ is the number of the last entry; $s$ is the shift of the number for the previous entry; $t$ is the time stamp corresponding to the last entry; $e_{1,m}, \ldots, e_{p,m}$ are the parameters of the last entry $\boldsymbol{e_m}$; $e_{1,m-s}, \ldots, e_{p,m-s}$ refer to parameters of the earlier received entry $\boldsymbol{e_{m-s}}$; $h_{1,m}, \ldots, h_{Q,m}$ are the hash values assigned by network verifiers to the last entry $\boldsymbol{e_m}$; $h_{1,m-s}, \ldots, h_{Q,m-s}$ are the hash values of the earlier received entry $\boldsymbol{e_{m-s}}$; and $q$ is the number of a verifier, $q = 1, \ldots, Q$. All these parameters are natural numbers. For simplicity, in [19] MVLL has used only the shift $s = 1$. Respectively, the truth table for MVLL can be given by Table 3.

**Table 3.** Truth table of the MVLL hash function $h^{(out)} = F_{ll}(\mathrm{m}, \mathrm{t}, \boldsymbol{e_m}, \boldsymbol{e_{m-1}}, \boldsymbol{h_m}, \boldsymbol{h_{m-1}})$ contains the newcomer entry $\boldsymbol{e_m} = (e_{1,m}, \ldots, e_{p,m})$, previous entry is $\boldsymbol{e_{m-1}} = \left(e_{1,m-1}, \ldots, e_{p,m-1}\right)$, and their sets of hash values $\boldsymbol{h_m} = (h_{1,m}, \ldots, h_{Q,m})$ and $\boldsymbol{h_{m-1}} = (h_{1,m-1}, \ldots, h_{Q,m-1})$, attributed to time moments $t_m$ and $t_{m-1}$.

| Input Variables | | | | | | | | | | | | | Output |
|---|---|---|---|---|---|---|---|---|---|---|---|---|---|
| **n** | **t** | $\mathbf{e_{1,t}}$ | $\ldots$ | $\mathbf{e_{p,t}}$ | $\mathbf{h_{1,t}}$ | $\ldots$ | $\mathbf{h_{q,t}}$ | $e_{1,t-1}$ | $\ldots$ | $e_{p,t-1}$ | $h_{1,t-1}$ | $\ldots$ | $\mathbf{h_{q,t-1}}$ | $h^{(out)}$ |
| 1 | $t_1$ | $e_{1,1}$ | $\ldots$ | $e_{p,1}$ | $h_{1,1}$ | $\ldots$ | $h_{Q,1}$ | $e_{1,0}$ | | $e_{p,0}$ | $h_{1,0}$ | $\ldots$ | $h_{Q,0}$ | $h^{(1,1)}$ |
| $\ldots$ | $\ldots$ | $\ldots$ | $\ldots$ | $\ldots$ | $\ldots$ | $\ldots$ | $\ldots$ | $\ldots$ | $\ldots$ | $\ldots$ | $\ldots$ | $\ldots$ | $\ldots$ | $\ldots$ |
| m-1 | $t_{m-1}$ | $e_{1,m-1}$ | $\ldots$ | $e_{p,m-1}$ | $h_{1,m-1}$ | $\ldots$ | $h_{Q,m-1}$ | $e_{1,m-2}$ | $\ldots$ | $e_{p,m-2}$ | $h_{1,m-2}$ | $\ldots$ | $h_{Q,m-2}$ | $h^{(m-1,1)}$ |
| m | $t_m$ | $e_{1,m}$ | $\ldots$ | $e_{p,m}$ | $h_{1,m}$ | $\ldots$ | $h_{Q,m}$ | $e_{1,m-1}$ | $\ldots$ | $e_{p,m-1}$ | $h_{1,m-1}$ | $\ldots$ | $h_{Q,m-1}$ | $h^{(m,1)}$ |

Equivalent representation of Table 3 by product terms is given by Exp. (6) based on definitions given in [50]:

$$
\begin{aligned}
h^{(out)} = \ & h^{(1,1)} \star X_m(1,1) \star X_t(t_1, t_1) \star X_{e,1,1}(e_{11}, e_{11}) \star \ldots \star X_{e,p,1}\left(e_{p,1}, e_{p,1}\right) \star X_{h,1,1}(h_{1,1}, h_{1,1}) \star \ldots \star \\
& X_{h,Q,1}\left(h_{Q,1}, h_{Q,1}\right) \star X_{e,1,0}(e_{1,0}, e_{1,0}) \star \ldots \star X_{e,p,0}\left(e_{p,0}, e_{p,0}\right) \star X_{h,1,0}(h_{1,0}, h_{1,0}) \star \ldots \star \\
& X_{h,Q,0}\left(h_{Q,m-1}, h_{Q,m-1}\right) + \ldots + h^{(m-1,1)} \star X_m(m-1, m-1) \star X_t(t_{m-1}, t_{m-1}) \star \\
& X_{e,1,m-1}(e_{1,m-1}, e_{1,m-1}) \star \ldots \star X_{e,p,m-1}\left(e_{p,m-1}, e_{p,m-1}\right) \star X_{h,1,m-1}(h_{1,m-1}, h_{1,m-1}) \star \ldots \star \\
& X_{h,Q,m-1}\left(h_{Q,m-1}, h_{Q,m-1}\right) \star X_{e,1,m-2}(e_{1,m-2}, e_{1,m-2}) \star \ldots \star X_{e,p,m-2}\left(e_{p,m-2}, e_{p,m-2}\right) \star \\
& X_{h,1,m-2}(e_{1,m-2}, e_{1,m-2}) \star \ldots \star X_{h,Q,m-2}\left(h_{Q,m-2}, h_{Q,m-2}\right) + h^{(m,1)} \star X_m(m, m) \star X_t(t_m, t_m) \star \\
& X_{e,1,m}(e_{1,m}, e_{1,m}) \star \ldots \star X_{e,p,m}\left(e_{p,m}, e_{p,m}\right) \star X_{h,1,m}(h_{1,m}, h_{1,m}) \star \ldots \star X_{h,Q,m}\left(h_{Q,m}, h_{Q,m}\right) \star \\
& X_{e,1,m-1}(e_{1,m-1}, e_{1,m-1}) \star \ldots \star X_{e,p,m-1}\left(e_{p,m-1}, e_{p,m-1}\right) \star X_{h,1,m-1}(h_{1,m-1}, h_{1,m-1}) \star \ldots \star \\
& X_{h,Q,m-1}\left(h_{Q,m-1}, h_{Q,m-1}\right).
\end{aligned}
\tag{6}
$$

### 2.4. Method of Additional Blocking Product Terms in the Linked List

This method [19] can prevent from attacks 2,3,6, based on adding of fake constants and excessive Literal parameters, given in Table 2. It uses the scheme to generate blocking product terms $PT^-$, providing subsuming of all illegally added product terms located in the truth table between any two given rows e.g., between entries $e_m$ and $e_{m-1}$. In order to apply it, the participants of the protocol should by default use constant $k-1$ only as the selected parameter for minimization procedures [18], which will always subsume all added product terms with any lower constants. For practice it is more comfort to represent steps

4 and 6, which were used in Algorithm 2 in the paper [19], as an Exp. (7) for i-th Literal, where indexed parameters $a, b$ are written as $\overline{x}_i, \overline{\overline{x}}_i$ for both entries:

$$
\begin{aligned}
&\widetilde{X}_i\left(\overline{x}_i, \overline{\overline{x}}\right) = \\
&\begin{cases}
X_i\left(MIN\left(\overline{x}_i, \overline{\overline{x}}_i\right)+1, Max\left(\overline{x}_i, \overline{\overline{x}}_i\right)-1\right), \ if \ Max\left(\overline{x}_i, \overline{\overline{x}}_i\right) - MIN\left(\overline{x}_i, \overline{\overline{x}}_i\right) \geq 2 \\
X_i\left(MIN\left(\overline{x}_i, \overline{\overline{x}}_i\right), Max\left(\overline{x}_i, \overline{\overline{x}}_i\right)\right), if \ Max\left(\overline{x}_i, \overline{\overline{x}}_i\right) - MIN\left(\overline{x}_i, \overline{\overline{x}}_i\right) < 2
\end{cases}
\end{aligned} \tag{7}
$$

As it is not correct to mix operators of AGA and Boolean logic [18], here operations MIN and MAX are considered as Boolean emulations of exp. (2) combined with adding or subtraction of 1. Then basing on definitions and Algorithm 2 [19] one can write logic expression, which combines two real product terms $PT_{m-1}^+$ and $PT_m^+$, responding to entries with logic constants equal to numbers $m$ and $m-1$, and the blocking term $PT_{m-1,m}^-$ with the logic constant $k-1$. Final exp. (8) obtained e.g., for 3 variables will be

$$
\begin{aligned}
&PT_{m-1}^+ + PT_{m-1,m}^- + PT_m^+ = \\
&= h_{m-1} \star X_1(\overline{x}_1, \overline{x}_1) \star X_2(\overline{x}_2, \overline{x}_2) \star X_3(\overline{x}_3, \overline{x}_3) + \\
&(k-1) \star \widetilde{X}_1\left(\overline{x}_1, \overline{\overline{x}}_2\right) \star \widetilde{X}_2\left(\overline{x}_1, \overline{\overline{x}}_2\right) \star \widetilde{X}_{e,1}\left(\overline{x}_1, \overline{\overline{x}}_2\right) + \\
&h_m \star X_1\left(\overline{\overline{x}}_1, \overline{\overline{x}}_1\right) \star X_2\left(\overline{\overline{x}}_2, \overline{\overline{x}}_2\right) \star X_3\left(\overline{\overline{x}}_3, \overline{\overline{x}}_3\right),
\end{aligned} \tag{8}
$$

where any $\widetilde{X}_i\left(\overline{x}_i, \overline{\overline{x}}\right)$ is given by Exp. (7).

As additional blocking segment $PT_{m-1,m}^-$ will enlarge the overall number of product terms and the computing time for ~50%, it may be reasonable to use it as a separate page (or a part) of the MVLL.

## 3. Results: Combined Hashing Scheme for MVL Linked List

In contrast to the paper [19], the modified MVLL scheme additionally uses:

(1) data, calculated by internal susbsystems of the agent, (2) binary *XOR* hashing of data of internal subsystems, and (3) selected parameters of external nodes.

### 3.1. Modified MVLL with Mixing Data from Internal Subsystems and External Nodes

The proposed scheme of data transfer for the network mobile agent is given in Figures 2 and 3, it involves partner agents $A_0$ and $A_1$ and $Q$ collaborating network nodes $V_q$, considered as verifiers $V$ like in [29,51]. It suppose three basic procedures I, II, and III marked by red, blue and green arrows. Steps I and II are shown in Figure 2, and step III is given in Figure 3.

The main task of step I in Figure 2 is to provide agent $A_1$ by the set of quasi-random keys by means QKD line, and to transform the finally processed quantum key into the subset of fragmented quasi-random keys written in subsystems $S_2, \ldots, S_N$. Thus, data flow I (red) supposes periodic loading of the mobile agent by quasi-random keys, received via the wireless QKD line [67,68], connecting agent $A_0$ with mobile agent $A_1$ in the trusted service zone providing appropriate level of security. The generated key is to be further fragmented to the set of 8-bit one-time quasi-random keys in order to adapt the system for 8-bit microcontrollers. These keys are to be accumulated and used for internal verification procedures, the needed throughput of quasi-random numbers source may be even less than $\sim 1$ Kbit/s. In the considered verification scheme the source of so-called entangled photons from route verification scheme [29] is not obligatory, and can be changed for any low-throughput QKD line, providing generation of quasi random one-time keys. For simplicity internal subsystem $S_1$ is shown as one supervisor module to control data exchange between external and internal devices.

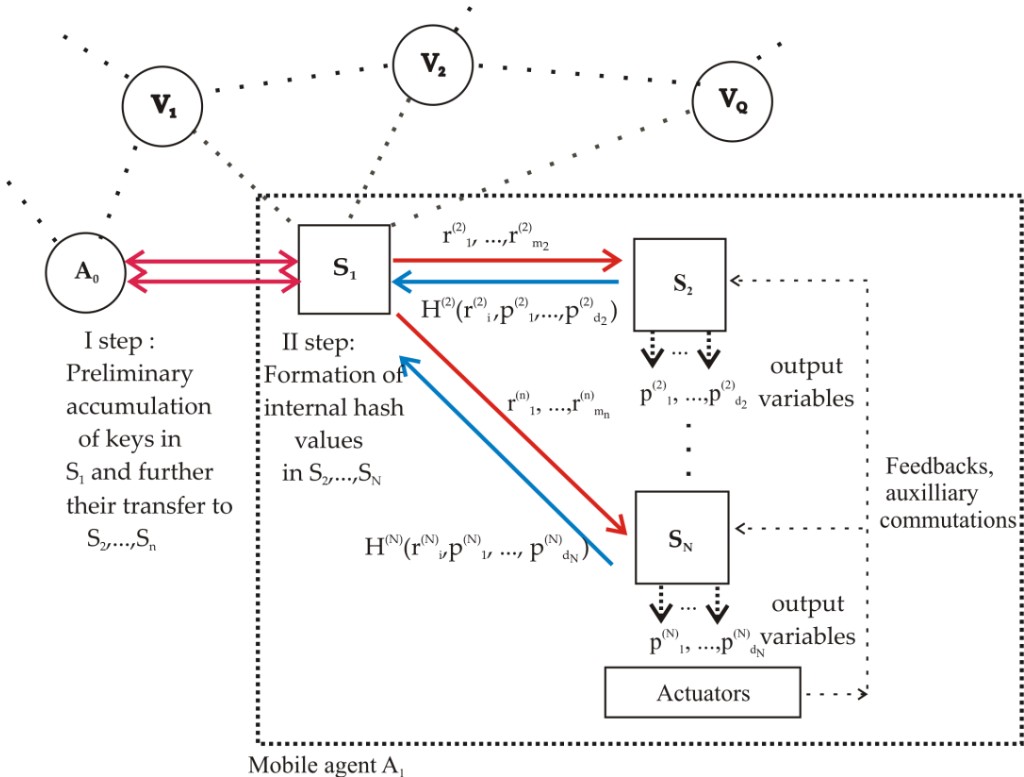

**Figure 2.** Steps I and II of the procedure to obtain the set of verification data for the joint linked list in the agent $A_1$, combining data of internal subsystems $p_1^{(n)}, \ldots, p_{d_N}^{(n)}$ and quasi-random keys $r_i^{(n)}$ b. y means of hashing functions $H^{(2)}, \ldots, H^{(N)}$.

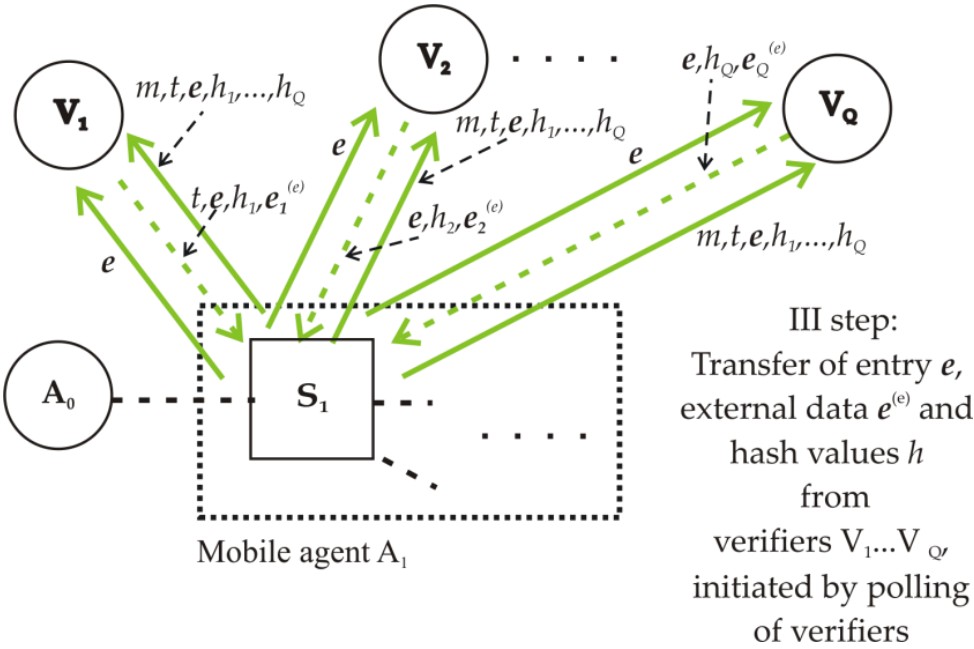

**Figure 3.** Step III of the procedure to obtain the set of verification data for the joint linked list. It discloses the interaction of agent $A_1$ with external verifiers $V_1$, $V_2$, $\ldots$, $V_Q$, combining data of internal subsystems and external verifiers.

When the set of shortened fragmentary quasi-random keys is formed, then the supervisor module $S_1$ chooses the time moment to generate the new entry for MVLL according

to the agent's task. This moment may e.g., respond to the visit to the next check-point. Then module $S_1$ activates the step II (tagged by blue arrows) to form the new entry, which includes the sequential polling of internal subsystems $S_2, \ldots, S_N$ to receive preliminary selected parameters for the external backup. Further internal hashing procedure should approve these data. Here for simplicity we do not consider any routine communications between the agent and external nodes which may interrupt the process.

Step III of the procedure is shown in Figure 3 and refers to the external distributed storage formation [19], which partially use blockchain scheme [25,26] with network protocols like e.g., [69,70]. In contrast to MVLL version proposed in [19], the new one may use more verification data from external nodes. E.g., space coordinates $(x, y, z)$ of the mobile checkpoint node can be useful to approve the passed route of the mobile agent. Polling of $Q$ free collaborating nodes (first green arrow) is to be done according to the initially given list of participants, and the reply transfer (backward oriented green arrow in middle) returns assigned quasi-random hash values and possible additional technical parameters. After the polling procedure the agent $A_1$ should resend the completed new entry to all involved participants (third green arrow). Data extraction from distributed external storages is supposed to be especially requested.

### 3.2. Possible Sources of Short Fragmented Keys

As QKD line can be blocked by intensive optical noize of any nature, alternative sources of quasi-random keys principally can be included into the scheme. These variants with lower level of privacy principally can be realized by the MVL–based scheme of the random oracle [57] or by a quantum random number generator [21]. Such alternatives does not disturb MVLL procedures.

Step I (red arrows) in Figure 2 basically suppose the periodical work of the wireless QKD line, providing quantum keys distribution for agents $A_0$ and $A_1$. The most secured level of QKD line is due to quantum mechanics and non-cloning theorem [61], which guarantees that if all basic protocols were carried out exactly, the generated key is known only to a pair of interacting abonents Alice and Bob with small enough probability of errors (see, e.g., review [12]). Modules Alice and Bob, see Figure 4a, use the conjugated pair of quantum and classical data lines and exploit specialized procedures, reglamented by cryptography standards by NIST [71], what limits the possibility to get inside and to add any additional verification procedures. Alice's and Bob's modules are to be integrated into internal subsystems of interacting agents, but MVLL procedures scarcely can be added into Bob module, and the separate verification subsystem $S_1$ is needed. As all certificated QKD lines provide at least NIST tests for randomness [71,72], the "block" test passing is guaranteed including 8- bit variant. Then the procedure for fragmenting of quantum data array into 8-bit fragmented keys can be choosen arbitrarily. Less protected versions, shown in Figure 4b, refer to alternative sources of quasi-random numbers like AGA-based random oracle [57] and QRNG [21].

### 3.3. Acquisition of Internal Parameters of the Agent and Their Hashing

Algorithm for acquisition of internal verification data is proposed in Algorithm 1. Every of subsystems modules $S_2, \ldots, S_N$ shown earlier in Figure 2 receives randomly given fragmented keys and uses them in calculation of hashing function $H^{(n)}(\mathbf{r}, \mathbf{p})$, where $\mathbf{p}$ includes current values of output variables of the subsystem $n$, and $r$ is a fragmented key. Components of $\mathbf{p}$ are preferably to be written in internal SRAM of a subsystem during the verification procedure, in order to be extracted in case of the detailed check. Coded by hashing procedure output variables are to be returned back from $S_2, \ldots, S_N$ to the verification module $S_1$, this process is shown by the blue arrows in Figure 2. Such mapping is to prevent direct illegal data reading from digital buses and is to check the operability of the subsystem in the coded form.

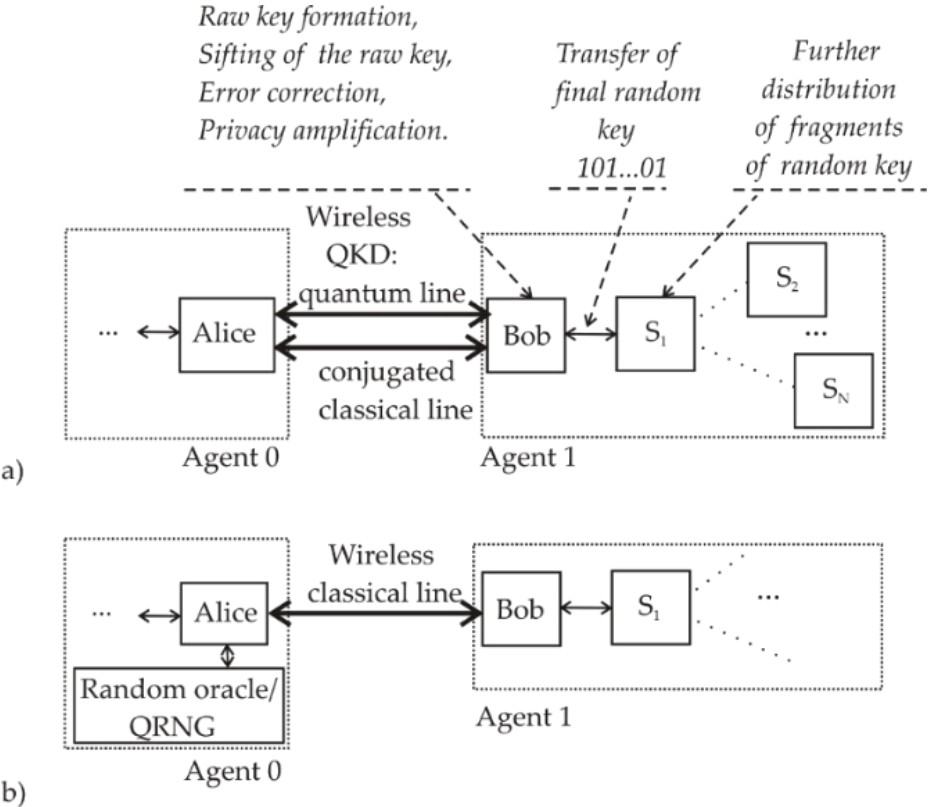

**Figure 4.** (**a**) Wireless QKD line can be regarded as the most secured source of quasi-random numbers for the pair of autonomous agents 0 and 1 of the MAS. (**b**) Less protected schemes may include QRNG and AGA-based random oracle.

According to Figure 2 MVLL is to include some variables of internal subsystems $S_2, \ldots, S_N$ and hash values assigned by $S_1$, optionally accompanied by some technical parameters of external nodes e.g., their coordinates. One can avoid here the direct use of internal time variable, as subsystems sequentially output values of variables $p_1^{(2)}, \ldots, p_{d_2}^{(2)}$, $\ldots, p_1^{(N)}, \ldots, p_{d_N}^{(N)}$ with much greater rate than external messages are received. Here for simplicity we suppose that all variables are calculated sequentially, where the sequence of operation responds to values $n = 1, 2 \ldots, N$, although the real time grid of their outputs can be more complex.

*3.4. Two-Steps XOR Hashing in Agent's Subsystems*

It seems more appropriate to use in internal subsystems (where possible!) simple and well-known binary logic hash functions in order to apply commercial modules and to shorten calculations. Principally, for hashing procedures one can design some MVL functions, but this seems to give advantages only in combination with specially planned AI procedures for subsystems, which are not discussed here. As it is being widely used in classic and quantum cryptography procedures [51], the well-known binary operator *XOR* ($\oplus$) [73,74] can be the first recommended variant for internal hashing, combining one-time quasi-random fragmented key $r_d^{(n)}$ and the set of $w_n$ output parameters $p_{w_n}^{(n)}$ of the subsystem $S_n$. Operator *XOR* is typically included into specifications of microcontrollers [74] and needs only 1 work cycle; it is oftenly tagged as *XRL* in microassembler specifications. Such two-stage hashing procedure is shown in the Table 4. The first step is to be calculated in internal subsystems $S_2, \ldots, S_N$. of the agent $A_1$. The second step of hashing is to be done in the supervisor module $S_1$ in order to minimize additional calculations in $S_2, \ldots, S_N$.

**Algorithm 1**: Acquisition of internal hashing parameters $H^{(n)}$ for $S_2, \ldots, S_N$.

| **Input:** | $k \leftarrow$ | Number of truth levels (not obligatory equal to 256); |
|---|---|---|
| | $N \leftarrow$ | Number of internal subsystems $S_1, \ldots, S_N$ of the agent $A_1$; |
| | $M \leftarrow$ | Maximal possible number of entries in MVLL; |
| | $d_n \leftarrow$ | Number of documented parameters of the subsystem, |
| | $w_2, \ldots, w_N \leftarrow$ | $d_n = 1, \ldots, D$; |
| | $r_1^{(2)}, \ldots, r_{d_2}^{(2)}$ | Numbers of output parameters in subsystems $S_2, \ldots, S_N$; |
| | $\quad \cdots \qquad \leftarrow$ | Sets of fragmented keys, preliminary prepared in subsystem $S_1$ |
| | $r_1^{(N)}, \ldots, r_{d_N}^{(N)}$ | for the new session of documentation in MVLL; |

| 1. | Subsystem $S_1$ | assigns counter $n = 2$; *As $S_1$ is the service subsystem!* |
|---|---|---|
| 2. | Subsystem $S_1$ | sets clocking signal ("write") to $S_n$ ; |
| | | transfer $r_{d_n}^{(n)}$ to subsystem $S_n$; |
| 3. | Subsystem $S_n$ | writes transferred $r_{d_n}^{(n)}$; |
| | | assigns $n = n + 1$; |
| 4. | Subsystem $S_1$ | checks if $n = N$, if yes goes to step 5, |
| | | otherwise goes to step 2; |
| | | calculates the set of all its output variables $\left\{ p_1^{(n)}, \ldots, p_{w_n}^{(n)} \right\}$; |
| | | writes $\left\{ p_1^{(n)}, \ldots, p_{w_n}^{(n)} \right\}$ in its internal memory; |
| 5. | Subsystem $S_n$ | calculates hash function $\boldsymbol{H}^{(n)}(r_{d_n}^{(n)}, p_1^{(n)}, \ldots, p_{d_n}^{(n)})$; |
| | | sets clocking signal ("write") to $S_1$ ; |
| | | transfer hash value $H(r_{d_n}^{(n)}, p_1^{(n)}, \ldots, p_{d_n}^{(n)})$ to $S_1$; |
| | | (*) *Hash function is discussed in Section 3.4* |
| | | (**) *Enlargement of n from 2 up to n responds to the basic sequence of* |
| | | *operation for subsystems in the agent $A_1$* |
| | | writes transferred $H^{(n)}(r_{d_n}^{(n)}, p_1^{(n)}, \ldots, p_{d_n}^{(n)})$; |
| 6. | Subsystem $S_1$ | assigns $n = n + 1$; |
| | | checks if $n = N$, if yes goes to Exit—step 7, |
| | | otherwise goes to step 5; |
| 7. | | Exit. |

| **Output:** $H^{(n)} \rightarrow$ | The set of data for the second step of hashing |
|---|---|

As it was shown in Section 2.2, several types of attacks aimed at MVL model can be based on modification of parameters of logic functions, written in the general format exp. (2) as arrays $A, B, C$ describing subsystems of the agent. That is why *XOR* hashing can be directly used for data protection of such arrays. This procedure is more visually shown in Figure 5, where quasi-random key $s$ $r_{d_n}^{(n)}$ may be either given as additional matrix elements in rows, or processed as additional separate column. Calculation of *XOR* hash values is firstly to be done for rows and then for column, it is the most simple and universal procedure to verify illegal modifications of the MVL function.

**Table 4.** Two steps of the internal hashing procedure of the agent $A_1$.

| | **Involved Subsystem** | **Expression** |
|---|---|---|
| I Step | | $H^{(n)}(r, \boldsymbol{p}) = r_d^{(n)} \oplus p_1^{(n)} \oplus p_2^{(n)} \oplus \ldots \oplus p_{w_n}^{(n)}$, $n = 2 \ldots, N$ |
| | Hash value for $S_2$ | $H^{(2)}(r, \boldsymbol{p})$, calculated in $S_2$ |
| | $\cdots$ | $\cdots$ |
| | Hash value for $S_N$ | $H^{(N)}(r, \boldsymbol{p})$, calculated in $S_N$ |
| II step | Integral hash value for all subsystems of the agent | $H^{(int)}(r, \boldsymbol{p}) ==$ $H^{(2)}(r, \boldsymbol{p}) H^{(3)}(r, \boldsymbol{p}) \ldots H^{(N)}(r, \boldsymbol{p})$ calculated in $S_1$ |

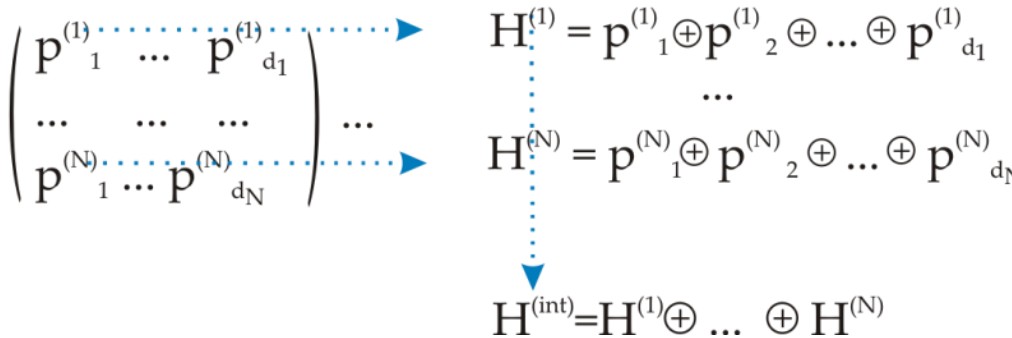

**Figure 5.** The general scheme to apply two step hashing to verification of MVL.

### 3.5. Formation of Modified MVLL with Internal and External Parameters

According to Figure 2 the entry for the modified version of MVLL should contain number $m$, time tag $t$, parameters $\boldsymbol{e}_m = \{e_{1,m}, \ldots, e_{p,m}\}$, and externally assigned hash values $h=\{h_1, \ldots, h_Q\}$. It also should include external and internal parameters $\boldsymbol{e}_m = \{\boldsymbol{e}_m^{(int)}, \boldsymbol{e}_m^{(ext)}\}$ instead of the set $\boldsymbol{e}_m = \{e_{1,m}, \ldots, e_{p,m}\}$, i.e., subsets $\boldsymbol{e}_m^{ext} = \{e_{1,m}^{ext}, \ldots, e_{v,m}^{ext}\}$ and $\boldsymbol{e}_m^{int} = \{e_{1,m}^{int}, \ldots, e_{p,m}^{int}\}$. For comfort we will use shortened notation $\boldsymbol{e}_m^{ext} = \boldsymbol{e}_m^{(e)}$ and $\boldsymbol{e}_m^{int} = \boldsymbol{e}_m$. Besides this, entry should contain internal hash functions $H^{(int)}(r, \boldsymbol{p})$ of the last and of the previous entry $\boldsymbol{e}_{m-1}$. The identifier of the agent and its license number are also actual parameters. Thus, modified MVLL function containing entries with numbers from 1 up to $m$ has the structure $h^{(out)} = F_{ll}(m, t, e_m, e_m^{(e)}, H_m^{(int)} h_{1,m}, \ldots, h_{Q,m}, e_{m-1}, e_{m-1}^{(e)}, H_{m-1}^{(int)} h_{1,m-1}, \ldots, h_{Q,m-1})$. Then final logic expression will respond to Exp. (9):

Note that data for external ledger should respond to 32–64 bit format of data, as internal microcontrollers or FPGA for IoT devices can mainly use 8-bit or sometimes 16-bit formats [75]. In any way, proposed formal procedures leave enough freedom of choice either to complement data by 0s for the necessary number of elder bits, or to sequentially unite bytes of several variables into one variable, or to combine both variants.

$$
\begin{aligned}
h^{(out)} = {}& \\
h^{(1,1)} &\star X_m(1,1) \star X_t(t_1, t_1) \star X_{e,1,1}(e_{11}, e_{11}) \star \ldots \star X_{e,p,1}(e_{p,1}, e_{p,1}) \\
&\star X_{e^{(e)},1,1}\left(e_{11}^{(e)}, e_{11}^{(e)}\right) \star \ldots \star X_{e^{(e)},v,1}\left(e_{v,1}^{(e)}, e_{v,1}^{(e)}\right) \star X_{h,1,1}(h_{1,1}, h_{1,1}) \star \ldots \\
&\star X_{h,Q,1}(h_{Q,1}, h_{Q,1}) \star X_{e,1,0}(e_{1,0}, e_{1,0}) \star \ldots \star X_{e,p,0}(e_{p,0}, e_{p,0}) \\
&\star X_{e^{(e)},1,0}\left(e_{10}^{(e)}, e_{10}^{(e)}\right) \star \ldots \star X_{e^{(e)},v,0}\left(e_{v,0}^{(e)}, e_{v,0}^{(e)}\right) \star X_{h,1,0}(h_{1,0}, h_{1,0}) \star \ldots \\
&\star X_{h,Q,0}(h_{Q,m-1}, h_{Q,m-1}) + \ldots + h^{(m-1,1)} \star X_m(m-1, m-1) \star X_t(t_{m-1}, t_{m-1}) \\
&\star X_{e,1,m-1}(e_{1,m-1}, e_{1,m-1}) \star \ldots \star X_{e,p,m-1}(e_{p,m-1}, e_{p,m-1}) \\
&\star X_{e^{(e)},1,m-1}\left(e_{1,m-1}^{(e)}, e_{1,m-1}^{(e)}\right) \star \ldots \\
&\star X_{e^{(e)},v,m-1}\left(e_{1,m-1}^{(e)}, e_{1,m-1}^{(e)}\right) \star X_{h,1,m}(h_{1,m-1}, h_{1,m-1}) \star \ldots \\
&\star X_{h,Q,m-1}(h_{Q,m-1}, h_{Q,m-1}) \star X_{e,1,m-2}(e_{1,m-2}, e_{1,m-2}) \star \ldots \\
&\star X_{e,p,m-2}(e_{p,m-2}, e_{p,m-2}) \star X_{e^{(e)},1,m-2}\left(e_{1,m-2}^{(e)}, e_{1,m-2}^{(e)}\right) \star \ldots \\
&\star X_{e^{(e)},v,m-2}\left(e_{v,m-2}^{(e)}, e_{v,m-2}^{(e)}\right) \star X_{h,1,m-2}(e_{1,m-2}, e_{1,m-2}) \star \ldots \\
&\star X_{h,Q,m-2}(h_{Q,m-2}, h_{Q,m-2}) + h^{(m,1)} \star X_m(m, m) \star X_t(t_m, t_m) \\
&\star X_{e,1,m}(e_{1,m}, e_{1,m}) \star \ldots \star X_{e,p,m}(e_{p,m}, e_{p,m}) \star X_{e^{(e)},1,m}\left(e_{1,m}^{(e)}, e_{1,m}^{(e)}\right) \star \ldots \\
&\star X_{e^{(e)},v,m}\left(e_{v,m}^{(e)}, e_{v,m}^{(e)}\right) \star X_{h,1,m}(h_{1,m}, h_{1,m}) \star \ldots \star X_{h,Q,m}(h_{Q,m}, h_{Q,m}) \\
&\star X_{e,1,m-1}(e_{1,m-1}, e_{1,m-1}) \star \ldots \star X_{e,p,m-1}(e_{p,m-1}, e_{p,m-1}) \\
&\star X_{e^{(e)},1,m-1}\left(e_{1,m-1}^{(e)}, e_{1,m-1}^{(e)}\right) \star \ldots \star X_{e^{(e)},v,m-1}\left(e_{v,m-1}^{(e)}, e_{v,m-1}^{(e)}\right) \\
&\star X_{h,1,m-1}(h_{1,m-1}, h_{1,m-1}) \star \ldots \star X_{h,Q,m-1}(h_{Q,m-1}, h_{Q,m-1})
\end{aligned}
\tag{9}
$$

## 4. Results: MVLL Scheme for Route Verification Task

As position verification methods may combine GPS, lidar, and computer vision systems, the route planning and the tracking estimation for mobile agents involves the problem of precision of data [76], influenced by the speed of mobile agents and satellites disposition. The controlled range for space coordinates can vary from centimeters up to thousands of km, and it is necessary to provide enough accuracy for all possible distances. First variant here is to apply correlated variables in the discrete *k*-valued logic model of AGA [18,19], where one can represent space coordinate *x* e.g., as 1560 m = 1 km + 500 m + 60 m+.... by the summation

$$x = x^{(1)} \times 1 + x^{(2)} \times 10 + \ldots x^{(p)} \times 10^p. \tag{10}$$

However, for the MVLL in route verification task there is the second possible scheme to apply purely formal representation excluding summation as in Exp. (13). It is based on the fact, that the distributed backup storage of MVLL should only measure and approve the correctness of data, as the value of coordinates discrepancy is required further at the stage of decision-making. Thus, verification or comparison of data from different storages mainly includes formal checks equal/non-equal, and further procedures are the another task. So that if unique GPS coordinates are represented [77] by a popular format, say, 57"45,4682, one can write it e.g., as the set of numbers {57}, {45}, {4682} = #{00111001}b, #{00101101}b, #{00010010; 01001010}b, fixing by default the position of separating tags. Advantage of formal AGA model is that for any bit format of space coordinates they can be quickly represented by several bytes in logic expressions with known by default rules for reverse reconstruction of geographical degrees, minutes, and seconds.

For the route vefification task the difference between the given and the real route should be finally described by the deviation function $F_{eval}$, which should estimate the difference between space coordinates $x, y$ for the planned route and measured coordinates $x^{(e)}, y^{(e)}$ : $F_{eval} = F(N_{agent}, N_{license}, x, y, x^{(e)}, y^{(e)})$. Close case here is the comparison of coordinates obtained by different GPS receivers, installed in navigation systems of the check-point and the mobile agent. Then the simplest way is to use MVLL with formal representation of coordinates. Realistic version of 8-bit MVLL seems to be represented by the entry described by Exp. (11)

$$e_m = \{m, t, ID_{agent}, N_{lic}, x^{(1)}, x^{(2)}, x^{(3)}, x^{(4)}, H^{(int)}, x^{(e1)}, x^{(e2)}, x^{(e3)}, x^{(e4)}, h_1, \ldots, h_Q\}, \tag{11}$$

where *m* is the entry number in the ledger, *t*—time of registration by the first of external verifying nodes, $ID_{agent}$—digital identifier of the agent (for simplicity we suppose it to be 1 byte), $N_{lic}$—license number (also 1 byte), $x^{(1)}, x^{(2)}, x^{(3)}, x^{(4)}$ describe 4 bytes of space coordinates, and data marked by $(^e)$ refer to another source of data, e.g., to the check-point. Here every set $x^{(1)}, x^{(2)}, x^{(3)}, x^{(4)}$ includes both the latitude and the longitude (N and E) in NMEA-0183 [78]). Variable $H^{(int)}$ in Exp. (11) describes values of internal hash function. Thus, entry $e_m$ in AGA can be written as the logic product term Exp. (12):

$$
\begin{aligned}
e_m = \\
h^{(m,1)} &\star X_m(m,m) \star X_t(t_m, t_m) \star X_{ID\,agent}(ID, ID) \star X_{lic}(N_{lic}, N_{lic}) \star X_{H^{(int)}}\left(H^{(int)}, H^{(int)}\right) \star \\
\star X_{x^{(1)}} &\left(x_m^{(1)}, x_m^{(1)}\right) \star X_{x^{(2)}}\left(x_m^{(2)}, x_m^{(2)}\right) \star X_{x^{(3)}}\left(x_m^{(3)}, x_m^{(3)}\right) \star X_{x^{(4)}}\left(x_m^{(4)}, x_m^{(4)}\right) \star \\
\star X_{x^{(e1)}} &\left(x_m^{(e1)}, x_m^{(e1)}\right) \star X_{x^{(e2)}}\left(x_m^{(e2)}, x_m^{(e2)}\right) \star X_{x^{(e3)}}\left(x_m^{(e3)}, x_m^{(e3)}\right) \star X_{x^{(e4)}}\left(x_m^{(e4)}, x_m^{(e4)}\right) \star \\
\star X_{h,1,m} &(h_{1,m}, h_{1,m}) \star \ldots \star X_{h,Q,m}(h_{Q,m}, h_{Q,m}) \star \\
\star X_{m-1} &(m-1, m-1) \star X_t(t_{m-1}, t_{m-1}) \star X_{ID\,agent}(ID, ID) \star X_{lic}(N_{lic}, N_{lic}) \star X_{H^{(int)}}\left(H^{(int)}, H^{(int)}\right) \star \\
\star X_{x^{(1)}} &\left(x_{m-1}^{(1)}, x_{m-1}^{(1)}\right) \star X_{x^{(2)}}\left(x_{m-1}^{(2)}, x_{m-1}^{(2)}\right) \star X_{x^{(3)}}\left(x_{m-1}^{(3)}, x_{m-1}^{(3)}\right) \star X_{x^{(4)}}\left(x_{m-1}^{(4)}, x_{m-1}^{(4)}\right) \star \\
\star X_{x^{(e1)}} &\left(x_{m-1}^{(e1)}, x_{m-1}^{(e1)}\right) \star X_{x^{(e2)}}\left(x_{m-1}^{(e2)}, x_{m-1}^{(e2)}\right) \star X_{x^{(e3)}}\left(x_{m-1}^{(e3)}, x_{m-1}^{(e3)}\right) \star X_{x^{(e4)}}\left(x_{m-1}^{(e4)}, x_{m-1}^{(e4)}\right) \star \\
\star X_{h,1,m-1} &(h_{1,m-1}, h_{1,m-1}) \star \ldots \star X_{h,Q,m-1}(h_{Q,m-1}, h_{Q,m-1}).
\end{aligned} \tag{12}
$$

Respectively, the resulting MVLL will be the set of *m*. product terms using pairs m/m-1 of entries given by Exp. (12) and combined by operators MAX from exp. (1). Principally, if a real system uses too many variables, it is possible to form one large scale MVLL as a common table of parameters, where several linked volumes or linked pages can be formed by some samplings of its variables in order to shorten the computing time for different procedures. Respectively, we has already mentioned above the possibility to use the set of blocking terms as a separate page of MVLL. Then such MVLL is to use several basic parameters $m, t, ID_{agent}, N_{license}$ for linking of pages. Another potential application for multi-page versions of MVLL is the detailed description of the history of routes by means of hash functions $h^{(m,m-s)}$ [19], where $s > 1$. But such versions of MVLL are out of the discussion here.

### 5. Results: MVLL in Microcontroller Module

As data verification in the hardware agent involves the local set of embedded microcontrollers [27], microassembler software is the priority tool for the design and debugging of such procedures. However, many popular platforms e.g., Arduino [79] are aimed at separate tasks with limited complexity and have too few parallel&serial ports for distributed schemes with AI procedures and flexible interaction of agent's subsystems. In contrast to them classical 8-bit microcontroller MCS-51 [74] is slow enough (up to 24–33 MHz), but provides greater fan-out due to its 4 independent parallel 8-bit banks including two serial ports. That is why MVLL software illustrations are shown further for the special dual-chip circuit board used in [19], which was proposed earlier for modelling of agent's AI functions and fuzzy logic, but which is also appropriate for modelling of MVLL. Such scheme is the way to design more complicated algorithms, as internal memory of MCS-51 can include only 500–600 microassembler instructions. As the set of embedded inctructions in MCS-51 typically provides very limited (64 kB) capacity of external SRAM, external trigger registers were used for the adressing of 1 MB external SRAM, what reduces time response but provides more flexibility.

Test programs for microcontrollers MCI and MCII were emulated by microassembler for MCS-51, simulator AVSIM and chip programmers KROT-MK and MASTER.

If the dual-chip module shown in Figure 6 is used as the model supervisor subsystem $S_1$, then signals coming from $S_2, \ldots, S_N$ can be loaded via input 8-bit bus connected to banks (or ports) P0 in both microcontrollers MCI and MCII. Respectively, output 8-bit bus can be involved for the transfer of data to subsystems $S_2, \ldots, S_N$ via MCIII and trigger registers Rg 4–7. In the dual-chip scheme microcontroller MCI is specialized for clocking of input signals, their procession, and read/write procedures; the second microcontroller MCII is mainly used for output signals control. Such dual-chip modules can be cascaded via digital buses.

Basic parameters were chosen as follows:

- $k$ = 256 logic levels for AGA model of MVLL,
- 8-bit format of data for internal microcontrollers,
- $D$ = 256 is the maximal number of documented entries in the MVLL,
- $N$ = 4 is the number of internal subsystems $S_2, \ldots, S_N$,
- $Q$ = 8 is the maximal number of external network verifiers.

Values of last two parameters were chosen to shorten the principal scheme.

The dual-chip scheme in Figure 6 contains a pair of 24 MHz ATMEL microcontrollers MCS-51 and has 512 KB ROM and 1 MB SRAM chips commutated to common 8 bit buses. ROM is principally necessary here to keep the backup data structure for data restoration in SRAM, ID identificator of the agent, license number, and the initial list of loyal network nodes. Trigger registers Rg1, 2, and 3 are used for addressing of SRAM, and $\overline{CE}, \overline{OE}, \overline{WE}$ are to control read/write procedures. Data from SRAM and ROM are to be loaded via banks P0 of MCI and MCII. Microcontroller MC1 has free pins in the port P3 for interaction with external decision-making module, which is out of discussion here. Port P3 in MCII is involved in clocking, e.g., pins P3.0-P3.4 control read/write procedures for $S_2, \ldots, S_4$.

Inverse signal $\overline{P3.0}$ initiates $S_2, \ldots, S_4$ to read the next fragmented 8-bit key, and $\overline{P3.1}$ sends them instruction to write the next hash value $H^{(int)}$. For larger $N$ one can use MCIII and additional registers.

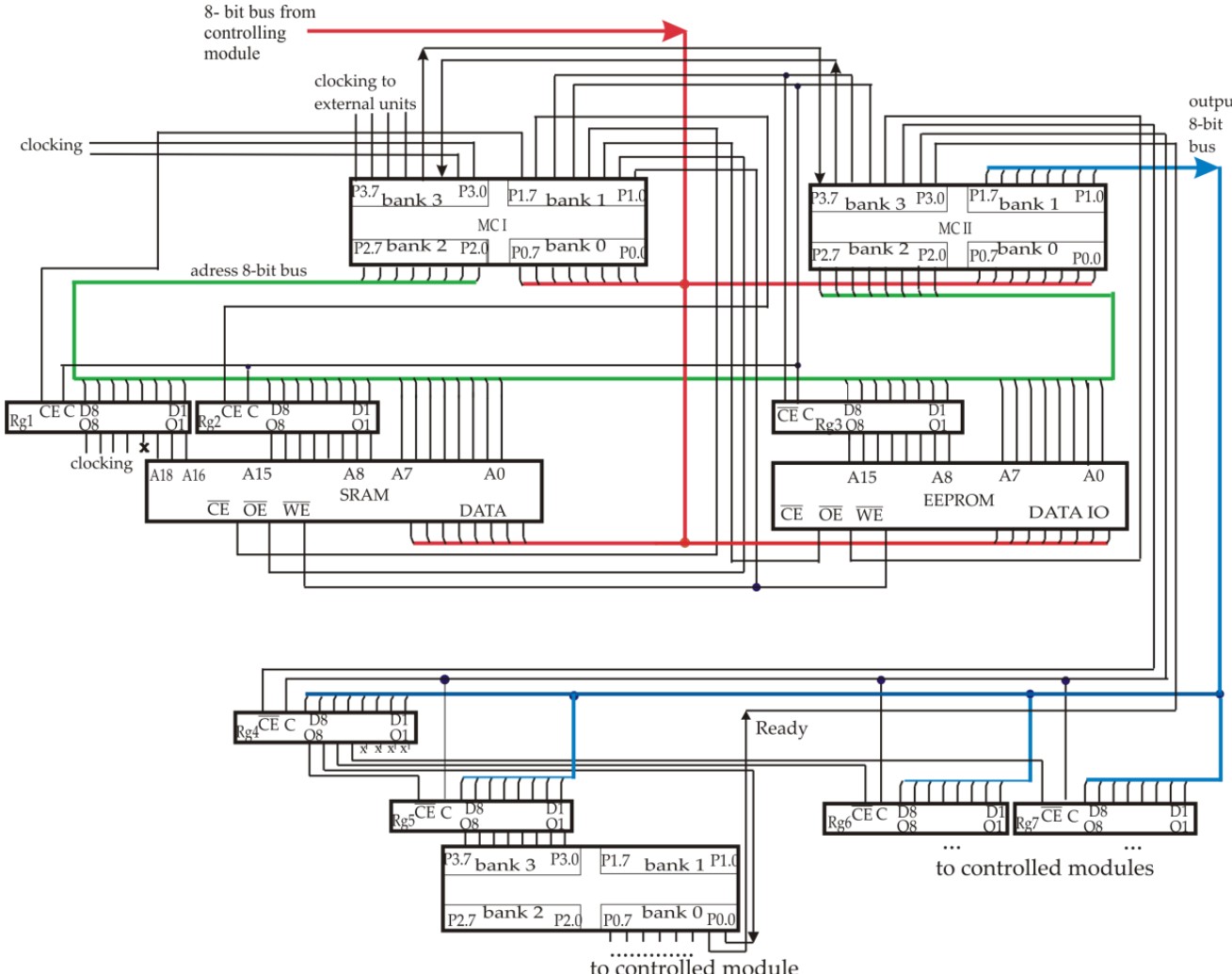

**Figure 6.** Version of two-chip circuit board [19] used for modelling of MVLL and interaction between subsystems $S_1$ and $S_2, \ldots,$ $Sn$ in the agent.

As an agent should process sensor data and transfer messages to external computers, chip MCIII in Figure 6 is to control peripheral modules, sensors, buffer SRAM, ADCs, DACs, step motors, and actuators. The circuit board shown in Figure 7 was used as the test subsystem with the serial link to PC.

Microcontroller MCS-51 can form and transfer data arrays with the length of 256 bytes accumulated in the buffer SRAM. Dual-chip scheme in Figure 6 is to be connected to the port P0 in MCS-51 in Figure 7 via MCIII, free ports P1-P2 commutated by the 8-bit data bus, and free pins. The model MVLL function was chosen as Exp. (13)

$$h^{(out)} =$$
$$F_{\text{ll}}(m,\ t,\ ID_{ag},\ N_{lic},\ x_m^{(1)}, x_m^{(2)}, x_m^{(3)}, x_m^{(4)}, H_m^{(int)}, x_m^{(e1)},\ x_m^{(e2)}, x_m^{(e3)}, x_m^{(e4)},$$
$$h_{1,m}, \ldots, h_{Q,m},\ x_{m-1}^{(1)}, x_{m-1}^{(2)},\ x_{m-1}^{(3)}, x_{m-1}^{(4)}, H_{m-1}^{(int)}, x_{m-1}^{(e1)},\ x_{m-1}^{(e2)}, x_{m-1}^{(e3)}, x_{m-1}^{(e4)},\ h_{1,m-1}, \ldots, h_{Q,m-1}) \tag{13}$$

where space coordinates were considered as formal non-correlated logic variables. In order to adapt the circuit boards to such MVLL, special SRAM structure should be used, which is briefly shown in Table 5 and is disclosed in details by Table 6.

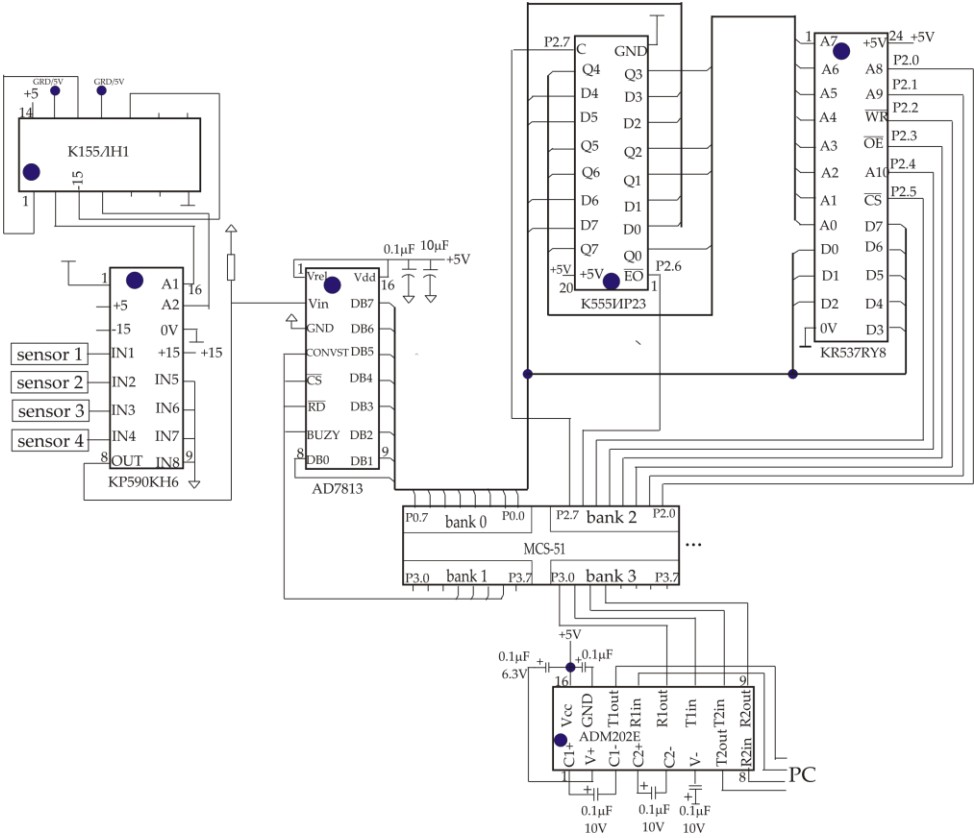

**Figure 7.** Circuit board with ADC AD7813, microcontroller MCS-51 and interface RS232C was used for imitation of subsystems $S_2, \ldots, S_4$. This module provides sensors data acquisition and their transfer to network PC via the serial port and ADM202E chip.

**Table 5.** Addresses in MVLL containing parameters $a, b$ of Literals.

| #A18-A16: | #000b | Internal | version | of entries | Parameter $a$ |
| #A18-A16: | #001b | Internal | version | of entries | Parameter $b$ |
| #A18-A16: | #100b | External | version | of entries | Parameter $a$ |
| #A18-A16: | #101b | External | version | of entries | Parameter $b$ |

Table 5 contains four equal parts. The first one refers to the internal version of entry $e_m$ and contains Literal's parameter. During initial formation of the MVLL from the given truth table (Table 1) namely equal values $a = b$ for Literals $X(a, b)$ are to be written in the MVLL. However, for practical applications the minimization may be used in order to shorten the calculation time. The efficiency of this procedure will depend on the specificity of data, and in the general case the number of entries $m$ may differ from the real number of product terms in MVLL. That is why one should obligatory reserve the second identical part of SRAM for Literal's parameters $b$, which can be activated by the increment of addresses #A18-A16 from 000 to 001. All other addresses will be the same. The third and the fourth parts of the SRAM refer to parameters $a$ and $b$ taken from the external request or verification task; they may be activated by #A18-A16=#100b and #101b. If one should approve only the presence of an external entry in the MVLL, this external entry is to be written to #A18-A16=#100b, but if one should approve the identity of external MVLL, both addresses #100b and #101b will be needed for parameters $a^{(e)}$ and $b^{(e)}$. Other free addresses can be involved for other external copies of MVLL.

**Table 6.** SRAM structure for MVLL segments with #A18-A16=#{000b, 001b, 100b, 101b}.

| Variable | $r_d^{(1)}$ | $r_d^{(2)}$ | $r_d^{(3)}$ | $r_d^{(4)}$ | $r_d^{(5)}$ | $r_d^{(6)}$ | $r_d^{(7)}$ | $r_d^{(8)}$ |
|---|---|---|---|---|---|---|---|---|
| Bytes/entry | 1 | 1 | 1 | 1 | 1 | 1 | 1 | 1 |
| Total, bytes | 2560 | 2560 | 2560 | 2560 | 2560 | 2560 | 2560 | 2560 |
| #A18-A16 | #000b | #000b | #000b | #000b | #000b | #000b | #000b | #000b |
| #SB | #0-9 | #10-19 | #20-29 | #30-39 | #40-49 | #50-59 | #60-69 | #70-79 |
| #LB | #0-255 | #0-255 | #0-255 | #0-255 | #0-255 | #0-255 | #0-255 | #0-255 |
| Variable | $h_1$ | $h_2$ | $h_3$ | $h_4$ | $h_5$ | $h_6$ | $h_7$ | $h^{(m,s)}$ |
| Bytes/entry | 1 | 1 | 1 | 1 | 1 | 1 | 1 | 1 |
| Total, bytes | 256 | 256 | 256 | 256 | 256 | 256 | 256 | 256 |
| #A18-A16 | #000b | #000b | #000b | #000b | #000b | #000b | #000b | #000b |
| #SB | #96 | #95 | #94 | #93 | #92 | #91 | #90 | #89 |
| #LB | #0-255 | #0-255 | #0-255 | #0-255 | #0-255 | #0-255 | #0-255 | #0-255 |
| Variable | $H^{(1)}$ | $H^{(2)}$ | $H^{(3)}$ | $H^{(4)}$ | $H^{(5)}$ | $H^{(6)}$ | $H^{(7)}$ | $H^{(8)}$ |
| Bytes/entry | 1 | 1 | 1 | 1 | 1 | 1 | 1 | 1 |
| Total, bytes | 256 | 256 | 256 | 256 | 256 | 256 | 256 | 256 |
| #A18-A16 | #000b | #000b | #000b | #000b | #000b | #000b | #000b | #000b |
| #SB | #88 | #87 | #86 | #85 | #84 | #83 | #82 | #81 |
| #LB | #0-255 | #0-255 | #0-255 | #0-255 | #0-255 | #0-255 | #0-255 | #0-255 |
| Variable | $x^{(1)}$ | $x^{(2)}$ | $x^{(3)}$ | $x^{(4)}$ | $x^{(e1)}$ | $x^{(e2)}$ | $x^{(e3)}$ | $x^{(e4)}$ |
| Bytes/entry | 1 | 1 | 1 | 1 | 1 | 1 | 1 | 1 |
| Total, bytes | 256 | 256 | 256 | 256 | 256 | 256 | 256 | 256 |
| #A18-A16 | #000b | #000b | #000b | #000b | #000b | #000b | #000b | #000b |
| #SB | #105 | #104 | #103 | #102 | #100 | #99 | #98 | #97 |
| #LB | #0-255 | #0-255 | #0-255 | #0-255 | #0-255 | #0-255 | #0-255 | #0-255 |
| Variable | $H_1^{(int)}$ | $H_2^{(int)}$ | $ID_{agent}$ | $N_{lic}$ | $m$ | $t$ | Counter | $w_n$ |
| Bytes/entry | 1 | 1 | 2 | 2 | 1 | 5 | 1 | 1 |
| Total, bytes | 256 | 256 | 2 | 2 | 256 | 25600 | 10 | 3 |
| #A18-A16 | #000b | #000b | #000b | #000b | #000b | #000b | #000b | #000b |
| #SB | #101 | #80 | #107 | #106 | #109 | #108 | #110-150 | #151 |
| #LB | #0-255 | #0-255 | #0-1 | #0-1 | #0-255 | #0-255 | #0-255 | #0-255 |
| Variable | $p_D^{(1)}$ | $p_D^{(2)}$ | $p_D^{(3)}$ | $p_D^{(4)}$ | $p_D^{(5)}$ | $p_D^{(6)}$ | $p_D^{(7)}$ | $p_D^{(8)}$ |
| Bytes/entry | 1 | 1 | 1 | 1 | 1 | 1 | 1 | 1 |
| Total, bytes | 256 | 256 | 256 | 256 | 256 | 256 | 256 | 256 |
| #A18-A16 | #000b | #000b | #000b | #000b | #000b | #000b | #000b | #000b |
| #SB | #152-161 | #162-171 | #172-181 | #182-191 | #192-201 | #202-211 | #212-221 | #222-231 |
| **#LB** | #0-255 | #0-255 | #0-255 | #0-255 | #0-255 | #0-255 | #0-255 | #0-255 |

Parts of MVLL (with #A18-A16=#{000b, 001b, 100b, 101b} has the structure, shown in Table 6. For every of entries $e_m$ besides its number $m$, $m \leq M = 256$, and the time stamp $t_m$, its SRAM segment contains the store of fragmented quasi-random keys $r_1^{(1)}, \ldots, r_D^{(8)}$, internal hash values $H^{(int)}$ (the second of them is the unused reserve for further) and $Q$ external hash values $h_1, \ldots, h_Q$. Also the entry $e_m$ should contain fields for agent's identifier $ID_{ag}$, license number $N_{lic}$. Special fields should be reserved for parameters $p_{d_n}^{(n)}$ received from internal subsystems $S_2, \ldots, S_4$. Besides this, auxiliary data fields in Counter can be used. Previous entry $e_{m-1}$ can be chosen by the input of $m - 1$.

The proposed MVLL structure saves enough unused cells and can be further optimized. One should especially note that as the register Rg1 in Figure 6 is used only partially, it potentially can address SRAM with 32 MB capacity for further AI modelling.

For software debugging it is more comfort to use the shortened and the regrouped fragment of SRAM shown as Table 7.

**Table 7.** Addresses #SB/LB of input variables and auxiliary counters in MVLL.

| Variable | m | t | $ID_{ag}$ | $N_{lic}$ | $x^{(1)}$ | $x^{(2)}$, | $x^{(3)}$ | $x^{(4)}$ |
|----------|---|---|-----------|-----------|-----------|------------|-----------|-----------|
| #SB | #109 | #108 | #107 | #106 | #105 | #104 | #103 | #102 |
| #LB | #0-255 | #0-255 | #0-255 | #0-255 | #0-255 | #0-255 | #0-255 | #0-255 |
| Variable | $H^{(int)}$ | $x^{(e1)}$ | $x^{(e2)}$, | $x^{(e3)}$ | $x^{(e4)}$ | $h_1$ | $h_2$ | $h_3$ |
| #SB | #101 | #100 | #99 | #98 | #97 | #96 | #95 | #94 |
| #LB | #0-255 | #0-255 | #0-255 | #0-255 | #0-255 | #0-255 | #0-255 | #0-255 |
| Variable | $h_4$ | $h_5$ | $h_6$ | $h_7$ | $h^{(out)}$ | - | - | - |
| #SB | #93 | #92 | #91 | #90 | #89 | - | - | - |
| #LB | #0-255 | #0-255 | #0-255 | #0-255 | #0-255 | - | - | - |
| Counter | $pt$ | $PT$ | $h^{(out)}$ | $h^{(*out)}$ | Identity | - | - | - |
| #SB | #110 | #111 | #112 | #113 | #114 | - | - | - |
| **#LB** | #0-255 | #0-255 | #0-255 | #0-255 | #0-255 | - | - | - |

Verification of data with the help of MVLL includes first of all: (1) processing of the data in the external node reply, sent to help the agent 1, and (2) reply of the subsystem $S_1$ to the external agent, requesting approval data for a third party.

The first case needs the simplest procedure if all the agents and nodes use by default the common format of MVLL. In fact it does not need any calculation of MVL functions and is the formal comparison of entry parameters, evaluated by one or the majority of data received from external nodes. The received data sequence refers to some entry $e_{m-k}$, where $k = 0, \ldots, M - 1$. As all addresses used in MVLL are fixed, the comparison of suspicious data and the received copy can be easily done by the cycle taken for variables from Table 7 and realized by the calculation of microassembler instruction CJNE A, P0, rel, where accumulator A is to be loaded by numbers received from external node, value of port $P0$ should be loaded from the internal copy of MVLL, and *rel* is the instruction to follow, if values of $A$ and $P0$ are not equal. Given further in Algorithm 2 software fragment CMPRE demonstrates this procedure. Cuts "iv" and "ev" tag internal and external variables. The number of external verifiers (loyal nodes) is 8 ones.

Application of this and given further subroutines suppose, that they are inserted after the instruction START into the microassembler set of instructions:

ORG 0H
AJMP START
ORG 30H
START: ...
... ...
END.

The second procedure mentioned above is to reply external request received from some other agent to approve entry $e_i^*$ for a third party. This case implies that $S_1$ in agent $A_1$ can process collective data within the ranks of collective collaboration of nodes. Then it is necessary to calculate MVLL function for the received external data and to check, if the external version is true. The task to calculate MVLL function $h^{(out)} = F_{ll}(e_i^{(*)})$ is disclosed in Algorithm 3, which is to use address data from Table 7. MVLL function is written via operators.

Literals ($X(a, b)$, MINs (*) and MAXs (+) given in Section 2, where any Literal $X(a, b)$ and product term *pt* can be equal either 0 or $k - 1$ only.

The first involved procedure is called SHPRTMS and is given by the subroutine in Algorithm 4. It calculates the set of shortened product terms $\{pt_1, \ldots, pt_m\}$ according to Algorithm 3 and procedures described in [50], using emulation of operators Literal and MIN given in [19].

---

**Algorithm 2.** Subroutine CMPRE is to compare the received external version of entry with the "suspicious" internal one. Both versions refer to the same agent.

---

**INPUT:** $h^{(out)} \leftarrow$ External parameter for approval,

$m, t, ID_{ag}, N_{lic}, x_m^{(1)}, \ldots, x_m^{(4)}, H_m^{(int)}, x_m^{(e1)}, \ldots, x_m^{(e4)}, h_{1,m}, \ldots, h_{8,m},$

$x_{m-1}^{(1)}, \ldots, x_{m-1}^{(4)}, H_{m-1}^{(int)}, x_{m-1}^{(e1)}, \ldots, x_{m-1}^{(e4)}, h_{1,m-1}, \ldots, h_{Q,m-1} \leftarrow$ Internal variables,

$m^{(*)}, t^{(*)}, ID_{ag}^{(*)}, N_{lic}^{(*)}, x_m^{(*1)}, \ldots, x_m^{(*4)}, H_m^{(*int)}, x_m^{(*e1)}, \ldots, x_m^{(*e4)}, h_{1,m}^{(*)}, \ldots, h_{8,m}^{(*)}$

$x_{m-1}^{(*1)}, \ldots, x_{m-1}^{(*4)}, H_{m-1}^{(*int)}, x_{m-1}^{(*e1)}, \ldots, x_{m-1}^{(*e4)}, h_{1,m-1}^{(*)}, \ldots, h_{8,m-1}^{(*)} \leftarrow$ External variables.

---

| | | | |
|---|---|---|---|
| 1 | CMPRE:MOV R7,#255; *counter entries m/m-1* | 2 | PRENT:MOV R3, R7; *fix #LB of entry* |
| 3 | MOV R6,#21; *count. of vars* | 4 | NEVAR:MOV P2, #0; *#A18-A16 =#0 for iv* |
| 5 | CLR P1.7; *enable Rg1by* $\overline{CE}$ | 6 | SETB P1.4; *write #0 to Rg1* |
| 7 | CLR P1.4 | 8 | SETB P1.7; *lock Rg1* |
| 9 | RDIV:MOV P2,#109; *#SB to read iv* | 10 | CLR P1.6; *enable Rg2 by* $\overline{CE}$ |
| 11 | SETB P1.4; *write #SB=#109 to Rg2* | 12 | CLR P1.4 |
| 13 | SETB P1.6; *lock Rg2* | 14 | MOV P2,#R3; *#LB=#255 for iv* |
| 15 | CLR P1.3; *enable SRAM by* $\overline{CE}$ | 16 | CLR P1.1; $\overline{OE}$ *enables output of iv* |
| 17 | MOV A,P0; *read iv to A* | 18 | SETB P1.1; *disable output of SRAM* |
| 19 | SETB P1.3; *disable SRAM by* $\overline{CE}$ | 20 | RDEV: *MOV P2,#100b; #A18-A16 for ev* |
| 21 | CLR P1.7; *enable Rg1by* $\overline{CE}$ | 22 | SETB P1.4; *write #SB=#109 to Rg1* |
| 23 | CLR P1.4 | 24 | SETB P1.7; *lock Rg1* |
| 25 | MOV R1,P0; *read ev* | 26 | CJNE A,R1,ERROR; *compare iv and ev* |
| 27 | DJNZ R6,NEVAR; *counter of vars* | 28 | ERROR: INC R5; *discrepancy counter* |
| 29 | CJNE R7,#253,PRENT; *process entry m-1* | 30 | RETI |

---

**OUTPUT:** R5 → Value of register R5 in MCS-51 indicates the number of discrepancies between internal and external versions of entry $e_i$(R5 = 0—equal, $\neq$0—number of errors).

---

**Algorithm 3:** The scheme to calculate $h^{(out)} = F_{ll}(e_i^{(*)})$ and to verify external entry.

---

| | |
|---|---|
| **Given external parameters:** | (1) vector of input variables $e_i$ combining entries $i$ and $i-1$; |
| | (2) declared hash value $h^{(out)}$ |

---

| | |
|---|---|
| 1. Calculation: | $h^{(out)} = F_{ll}(e_i) = h^{(1,1)} * pt_1(e_i) + \ldots + h^{(m,1)} * pt_m(e_i)$, where $pt_k(x) = X_1 * \ldots * X_Y$ tags the shortened product term containing only Literals and MINs (*) without constants $h^{(i,1)}$. **Procedure includes 3 subroutines:** |
| Non-zero *pts*: | $\{pt_1, \ldots, pt_m\}$; #SB=#110 |
| Set of MINs: | $PT_1 = MIN(h^{(1,1)}, pt_1), \ldots, PT_m = MIN(h^{(m,1)}, pt_m)$; #SB=#111 |
| MAX: | $h^{(ll)} = \text{MAX}(PT_1, \ldots, PT_m)$; #SB=#112 |
| 2. Comparison: | of declared external $h^{(*out)}$ with internal $h^{(out)}$; #SB=#113-114 |

---

**Algorithm 4:** Subroutine SHPRTMS is to calculate shortened product terms $\{pt_1, \ldots, pt_m\}$.

---

**INPUT:** $m, t, ID_{ag}, N_{lic}, x_m^{(1)}, x_m^{(2)}, x_m^{(3)}, x_m^{(4)}, H_m^{(int)}, x_m^{(e1)}, x_m^{(e2)}, x_m^{(e3)}, x_m^{(e4)},$

$h_{1,m}, \ldots, h_{Q,m} \leftarrow$ Input internal variables

---

| | | | |
|---|---|---|---|
| 1 | SHPRMS: MOV R4,#255; *counter of pts* | 2 | MOV R3, #109; *#SB-counter of vars* |
| 3 | NPT: MOV R2,#255; *counter of entries* | 4 | NVAR1: MOV P2,#000b; *#A18-A16=#000b for a* |
| 5 | CLR P1.7; *enable Rg1 by* $\overline{CE}$ | 6 | SETB P1.4; *Rg1 writes #A18-A16* |
| 7 | CLR P1.4; | 8 | SETB P1.7; *lock Rg1* |
| 9 | MOV P2,R3; *#SB=#109* | 10 | CLR P1.6; *enable Rg2 by* $\overline{CE}$ |
| 11 | SETB P1.4; *write #SB=#109 to Rg2* | 12 | CLR P1.4 |
| 13 | SETB P1.6; *lock Rg2* | 14 | MOV P2,R2; *#LB counter of entries* |
| 15 | CLR P1.3; *enable SRAM by* $\overline{CE}$ | 16 | CLR P1.1; $\overline{OE}$ *enables output of SRAM* |
| 17 | REAI:MOV R7,P0; *read a* | 18 | SETB P1.1; *disable output of SRAM* |
| 19 | SETB P1.3; *disable SRAM* | 20 | MOV P2,#001b; *#A18-A16=#001b for b* |

| | | | | |
|---|---|---|---|---|
| 21 | CLR P1.7; *enable Rg1 by $\overline{CE}$* | 22 | SETB P1.4; *Rg1 writes #A18-A16* |
| 23 | CLR P1.4; | 24 | SETB P1.7; *lock Rg1* |
| 25 | MOV P2,R3; *#SB=#109* | 26 | CLR P1.6; *enable Rg2 by $\overline{CE}$* |
| 27 | SETB P1.4; *write #SB to Rg2* | 28 | CLR P1.4 |
| 29 | SETB P1.6; *lock Rg2* | 30 | MOV P2,R2; *#LB counter of entries* |
| 31 | CLR P1.3; *enable SRAM by $\overline{CE}$* | 32 | CLR P1.1; *$\overline{OE}$ enables output of SRAM* |
| 33 | REBI:MOV R6,P0; *read b* | 34 | SETB P1.1; *disable output of SRAM* |
| 35 | SETB P1.3; *disable SRAM* | 36 | MOV P2,#100b; *ext var adress is #100b* |
| 37 | CLR P1.7; *enable Rg1 by $\overline{CE}$* | 38 | SETB P1.4; *Rg1 writes #A18-A16* |
| 39 | CLR P1.4; | 40 | SETB P1.7; *lock Rg1* |
| 41 | MOV P2,R3; *#SB* | 42 | CLR P1.6; *enable Rg2 by $\overline{CE}$* |
| 43 | SETB P1.4; *write #SB to Rg2* | 44 | CLR P1.4 |
| 45 | SETB P1.6; *lock Rg2* | 46 | MOV P2,R2; *#LB counter of entries* |
| 47 | CLR P1.3; *enable SRAM by $\overline{CE}$* | 48 | CLR P1.1; *$\overline{OE}$ enables output of SRAM* |
| 49 | REVAR:MOV R5,P0; *read input var* | 50 | SETB P1.1; *disable output of SRAM* |
| 51 | SETB P1.3; *disable SRAM* | 52 | LITE: MOV A,R7; *load a to calc Lit* |
| 53 | CLR C; *prepare carry bit* | 54 | SUBB A, R5; *a-var* |
| 55 | JC CAB; *jump if carry bit C=1 and varGRa* | 56 | AJMP PT0; *Lit=0 and the whole pt=0* |
| 57 | CAB: CLR C | 58 | MOV A,R6; *load b to calc Lit* |
| 59 | CLR C; *prepare carry bit* | 60 | SUBB A,R5; *b-var* |
| 61 | JC PT0; *go PT0 if bit C=1 as var GR b* | 62 | NOP |
| 63 | DEC R3; | 64 | CJNE R3,#90,NVAR1; |
| 65 | MOV R3,#109; *#SB in previous entry* | 66 | DEC R2; *for previous entry #LB=254* |
| 67 | NVAR2:MOV P2,#000b; *#000b for a* | 68 | CLR P1.7; *enable Rg1 by $\overline{CE}$* |
| 69 | SETB P1.4; *Rg1 writes #A18-A16* | 70 | CLR P1.4; |
| 71 | SETB P1.7; *lock Rg1* | 72 | MOV P2,R3; *write #SB* |
| 73 | CLR P1.6; *enable Rg2 by $\overline{CE}$* | 74 | SETB P1.4; *write #SB=#109 to Rg2* |
| 75 | CLR P1.4 | 76 | SETB P1.6; *lock Rg2* |
| 77 | MOV P2,R2; *#LB counter of entries* | 78 | CLR P1.3; *enable SRAM by $\overline{CE}$* |
| 79 | CLR P1.1;*$\overline{OE}$ enables output of SRAM* | 80 | REAP:MOV R7,P0; *read a* |
| 81 | SETB P1.1; *disable output of SRAM* | 82 | SETB P1.3; *disable SRAM* |
| 83 | MOV P2,#001b; *#A18-A16=#001b for b* | 84 | CLR P1.7; *enable Rg1 by $\overline{CE}$* |
| 85 | SETB P1.4; *Rg1 writes #A18-A16* | 86 | CLR P1.4; |
| 87 | SETB P1.7; *lock Rg1* | 88 | MOV P2,R3; *#SB* |
| 89 | CLR P1.6; *enable Rg2 by $\overline{CE}$* | 90 | SETB P1.4; *write #SB to Rg2* |
| 91 | CLR P1.4 | 92 | SETB P1.6; *lock Rg2* |
| 93 | MOV P2,R2; *#LB counter of entries* | 94 | CLR P1.3; *enable SRAM by $\overline{CE}$* |
| 95 | CLR P1.1; *$\overline{OE}$ enables output of SRAM* | 96 | REBP:MOV R6,P0; *read b* |
| 97 | SETB P1.1; *disable output of SRAM* | 98 | SETB P1.3; *disable SRAM* |
| 99 | MOV P2,#100b; *#A18-A16=#011b for b* | 100 | CLR P1.7; *enable Rg1 by $\overline{CE}$* |
| 101 | SETB P1.4; *Rg1 writes #A18-A16* | 102 | CLR P1.4; |
| 103 | SETB P1.7; *lock Rg1* | 104 | MOV P2,R3; *write in #SB=#109* |
| 105 | CLR P1.6; *enable Rg2 by $\overline{CE}$* | 106 | SETB P1.4; *write #SB to Rg2* |
| 107 | CLR P1.4 | 108 | SETB P1.6; *lock Rg2* |
| 109 | MOV P2,R2; *#LB counter of entries* | 110 | CLR P1.3; *enable SRAM by $\overline{CE}$* |
| 111 | CLR P1.1; *$\overline{OE}$ enables output of SRAM* | 112 | REVAR2:MOV R5,P0; *read inp var* |
| 113 | SETB P1.1; *disable output of SRAM* | 114 | SETB P1.3; *disable SRAM* |
| 115 | LITPE: MOV A,R7; *load a for Lit* | 116 | CLR C; *prepare carry bit* |
| 117 | SUBB A, R5; *a-var* | 118 | JC CMB2; *go CMB2 if C=1 as var GR a* |
| 119 | AJMP PT0; *Lit = 0 and whole pt = 0* | 120 | CMB2: CLR C; |
| 121 | MOV A,R6; *load b to compare with var* | 122 | CLR C; *prepare carry bit* |
| 123 | SUBB A,R5; *b-var* | 124 | JC PT0; *go PT0 if C = 1 as var GR b* |
| 125 | DEC R3; | 126 | CJNE R3,#90, NVAR2; |
| 127 | PT0: MOV R0, #0; *counter = #0* | 128 | AJMP WRLITER; |
| 129 | PT1: MOV R0, #255; *counter = #255* | 130 | WRLITER:MOV P2,#000b; *write results* |
| 131 | CLR P1.7; *enable Rg1 by $\overline{CE}$* | 132 | SETB P1.4; *Rg1 writes #A18-A16* |
| 133 | CLR P1.4; | 134 | SETB P1.7; *lock Rg1* |

| | | | |
|---|---|---|---|
| 135 | MOV P2,#110; *to write in #SB=#110* | 136 | CLR P1.6; *enable Rg2 by $\overline{CE}$* |
| 137 | SETB P1.4; *write #SB=#110 to Rg2* | 138 | CLR P1.4 |
| 139 | SETB P1.6; *lock Rg2* | 140 | MOV P2,R2; *#LB is the counter of pt* |
| 141 | CLR P1.3; *enable SRAM by $\overline{CE}$* | 142 | CLR P1.1;*$\overline{OE}$ enables output of SRAM* |
| 143 | MOV P0,R0; *write result from R0* | 144 | SETB P1.1; *disable output of SRAM* |
| 145 | SETB P1.3; *disable SRAM* | 146 | DJNZ R4,NPT; *test new pt* |
| 147 | RETI | 148 | |

**OUTPUT:**$\{pt_1, \ldots, pt_m\} \rightarrow$ "Shortened" product terms are written in SRAM

Second subroutine MINHMPT is given in Algorithm 5 and calculates the set of "full" product terms including logic constants $h^{(i,1)}$, it exploits only operators MIN.

---

**Algorithm 5:** Subroutine MINHMPT to calculate the set of full product terms $PT_1 = MIN\left(h^{(1,1)}, pt_1\right), \ldots, PT_m = MIN(h^{(m,1)}, pt_m)$.

**INPUT:** $pt_1, \ldots, pt_m \leftarrow$ "Shortened" product terms

| | | | |
|---|---|---|---|
| 1 | MINHMPT:MOV R1, #89; *#SB to $h^{(i,1)}$* | 2 | MOV R2, #255; *counter of pts* |
| 3 | MOV R3, #110; *#SB=#110 to read pts* | 4 | NEXTPT:MOV P2,#000b; *#A18-A16* |
| 5 | CLR P1.7; *enable Rg1 by $\overline{CE}$* | 6 | SETB P1.4; *Rg1 writes #A18-A16* |
| 7 | CLR P1.4 | 8 | SETB P1.7; *lock Rg1* |
| 9 | MOV P2,R1; *addressing #SB* | 10 | CLR P1.6; *enable Rg2by $\overline{CE}$* |
| 11 | SETB P1.4; *write #SB to Rg2* | 12 | CLR P1.4 |
| 13 | SETB P1.6; *lock Rg2* | 14 | MOV P2,R2; *#LB is the counter of pts* |
| 15 | CLR P1.3; *enable SRAM by $\overline{CE}$* | 16 | CLR P1.1;*$\overline{OE}$ enables output of SRAM* |
| 17 | MOV A,P0; *read h* | 18 | SETB P1.1; *disable output of SRAM* |
| 19 | SETB P1.3; *disable SRAM* | 20 | MOV P2,R3; *#SB to read pt* |
| 21 | CLR P1.6; *enable Rg2by $\overline{CE}$* | 22 | SETB P1.4; *write #SB to Rg2* |
| 23 | CLR P1.4 | 24 | SETB P1.6; *lock Rg2* |
| 25 | CLR P1.3; *enable SRAM by $\overline{CE}$* | 26 | CLR P1.1;*$\overline{OE}$ enables output of SRAM* |
| 27 | MOV R7,P0; *read pt* | 28 | SETB P1.1; *disable output of SRAM* |
| 29 | SETB P1.3; *disable SRAM* | 30 | MOV R3,A; *save value of Acc* |
| 31 | CLR C | 32 | SUBB A,R7 |
| 33 | JNC MIN_H | 34 | MOV R0,R7 |
| 35 | MIN_H:MOV R0,R3; *MIN value is in R0* | 36 | WRMINPT:MOV P2,#000b; *#A18-16* |
| 37 | CLR P1.7; *enable Rg1 by $\overline{CE}$* | 38 | SETB P1.4; *Rg1 writes#A18-A16* |
| 39 | CLR P1.4; | 40 | SETB P1.7; *lock Rg1* |
| 41 | MOV P2,#111; *write in #SB=#111* | 42 | CLR P1.6; *enable Rg2by $\overline{CE}$* |
| 43 | SETB P1.4; *write #SB=#111 to Rg2* | 44 | CLR P1.4 |
| 45 | SETB P1.6; *lock Rg2* | 46 | MOV P2,R2; *#LB is the counter of pts* |
| 47 | CLR P1.3; *enable SRAM by $\overline{CE}$* | 48 | CLR P1.1;*$\overline{OE}$ enables output of SRAM* |
| 49 | MOV P0,R0; *write result of comparison* | 50 | SETB P1.1; *disable output of SRAM* |
| 51 | SETB P1.3; *disable SRAM* | 52 | DJNZ R2, NEXTPT |
| 53 | RETI | | |

**OUTPUT:** $PT_1, \ldots, PT_m \rightarrow$ 256 "full" product terms are written in SRAM

---

The first of necessary subroutines is the third MVLL procedure MAXPT uses operator MAX for the set of obtained above product terms PTs and is given in Algorithm 6.

The obtained value $h^{(out)}$ is to be compared with the declared external version $h^{(*out)}$; this should approve (or not) data of the third party.

Given above result has used internal hash values $H^{(int)}$, which can be calculated by means of two simple hashing procedures for binary *XOR* given above in Section 3.4. This method of hashing is represented by subroutines HASHN (I step) and HASHINT (II step). They directly respond to procedures from Algorithm 3 and are shown in Algorithms 7 and 8. The first of them estimates the hash value for quasi-random fragmented key and internal parameters. The used for hashing binary *XOR* operator responds in the specification list of

MCS-51 to *XRL*. Name of instruction *RDWN* prompts the reading of corresponding $w_n$, as in general case the number of documented parameters of subsystems may differ.

---

**Algorithm 6:** Subroutine MAXPTS calculates final result $PT_m = \text{MAX}\,(PT_1, \dots, PT_m)$.

**INPUT:** $PT_1, \dots, PT_m \leftarrow$ "Full" product terms

| | | | | |
|---|---|---|---|---|
| 1 | MAXPTS:MOV R1,#111; *#SB111 for PTs* | 2 | MOV R2,#255; *counter of PTs* |
| 3 | MOV P2,#000b; *output #A18-A16 for Rg1* | 4 | CLR P1.7; enable Rg1 by $\overline{CE}$ |
| 5 | SETB P1.4; *Rg1 writes #A18-A16* | 6 | CLR P1.4 |
| 7 | SETB P1.7; *lock Rg1* | 8 | MOV P2,R1; *addressing #SB=#111* |
| 9 | CLR P1.6; *enable Rg2 by* $\overline{CE}$ | 10 | SETB P1.4; *write #SB=#111 to Rg2* |
| 11 | CLR P1.4 | 12 | SETB P1.6; *lock Rg2* |
| 13 | MOV P2,R2; *#LB is the counter of PTs* | 14 | CLR P1.3; *enable SRAM by* $\overline{CE}$ |
| 15 | CLR P1.1;$\overline{OE}$ *enables output of SRAM* | 16 | MOV A,P0; *read PT* |
| 17 | SETB P1.1; *disable output of SRAM* | 18 | SETB P1.3; *disable SRAM* |
| 19 | DEC R2 | 20 | NEXTPT:MOV P2,R2 |
| 21 | CLR P1.3; *enable SRAM by* $\overline{CE}$ | 22 | CLR P1.1;$\overline{OE}$ *enables output of SRAM* |
| 23 | MOV R7,P0; *read next PT* | 24 | SETB P1.1; *disable output of SRAM* |
| 25 | SETB P1.3; *disable SRAM* | 26 | MOV R3,A; *save value of Acc* |
| 27 | CLR C | 28 | SUBB A,R7 |
| 29 | JNC MAX_N1 | 30 | MOV R0,R7 |
| 31 | MAX_N1:MOV R0,R3 | 32 | DJNZ R2, NEXTPT |
| 33 | RETI | | |

**OUTPUT**: R0 $\rightarrow h^{(out)}$ , result is written in register R0

---

**Algorithm 7:** Subroutine **HASHN** (I step).

INPUT: $N \leftarrow$ number of subsystems (N = 3)

$w_1, \dots, w_n \leftarrow$ numbers of documented parameters in sybsystems

$r_{d_n}^{(n)} \leftarrow$ quasi-random fragmented keys

| | | | | |
|---|---|---|---|---|
| 1 | HASHN: MOV R3,#255; *initial #LB* | 2 | MOV R2,#3; *N of subsystems* |
| 3 | RDWN:MOV P2,#151; *SB to read WN* | 4 | CLR P1.6; *enable Rg2by* $\overline{CE}$ |
| 5 | SETB P1.4; *write #SB=#151 to Rg2* | 6 | CLR P1.4 |
| 7 | SETB P1.6; *lock Rg2* | 8 | MOV P2,#R3; *addressing #LB* |
| 9 | CLR P1.3; *enable SRAM by* $\overline{CE}$ | 10 | CLR P1.1; $\overline{OE}$ *enables output WN* |
| 11 | MOV A,P0; *read WN* | 12 | SETB P1.1; *disable SRAM output* |
| 13 | SETB P1.3; *disable SRAM* | 14 | RDLB: MOV P2,#10; *#SB of rd2* |
| 15 | CLR P1.6; *enable Rg2by* $\overline{CE}$ | 16 | SETB P1.4; *write #SB to Rg2* |
| 17 | CLR P1.4 | 18 | SETB P1.6; *lock Rg2* |
| 19 | NEXTKEY1: MOV P2,R3; *#LB* | 20 | CLR P1.3; *enable SRAM by* $\overline{CE}$ |
| 21 | CLR P1.1;$\overline{OE}$ *enables output of LB* | 22 | MOV R4,P0; *read current LB* |
| 23 | SETB P1.1; *disable output of SRAM* | 24 | SETB P1.3; *disable SRAM* |
| 25 | NEXTHN:MOV R4, #87; *go to # of H(2)* | 26 | RDRD2: MOV P2,R4; *read H(2)* |
| 27 | CLR P1.6; *enable Rg2by* $\overline{CE}$ | 28 | SETB P1.4; *write #87 to Rg2* |
| 29 | CLR P1.4 | 30 | SETB P1.6; *lock Rg2* |
| 31 | NEXTKEY2: MOV P2,R3; *addressing #LB* | 32 | CLR P1.3; *enable SRAM by* $\overline{CE}$ |
| 33 | CLR P1.1;$\overline{OE}$ *enables output rd* | 34 | MOV A,P0; *read current rd* |
| 35 | RDPN: MOV P2,#162; *#SB(Rg2) to read pn* | 36 | CLR P1.6; *enable Rg2by* $\overline{CE}$ |
| 37 | SETB P1.4; *write #162 to Rg2* | 38 | CLR P1.4 |
| 39 | SETB P1.6; *lock Rg2* | 40 | NEXTKEY: MOV P2,R3; *#LB* |
| 41 | CLR P1.3; *enable SRAM by* $\overline{CE}$ | 42 | CLR P1.1;$\overline{OE}$*enables output p2* |
| 43 | MOV R1,P0; *read current pn* | 44 | XOR:XRL A,R1; *rd XOR p2 write* |
| 45 | DJNZ R3,RDPN | 46 | MOV R5,#89; *go field of H(n)* |
| 47 | WRHN: MOV P2,R5; *SB to write H(n)* | 48 | CLR P1.6; enable Rg2by $\overline{CE}$ |

| 49 | SETB P1.4; *write #89 to Rg2* | 50 | CLR P1.4 |
| 51 | SETB P1.6; *lock Rg2* | 52 | MOV P2,R3; *#LB to write H(n)* |
| 53 | CLR P1.3; *enable SRAM by $\overline{CE}$* | 54 | CLR P1.0; *enable $\overline{WR}$ of H(n)* |
| 55 | MOV P0,A; *write H(n) from ACC* | 56 | SETB P1.0; *disable $\overline{WR}$* |
| 57 | SETB P1.3; *disable SRAM* | 58 | DJNZ R2,XOR; *next n+1 for H(n)* |
| 59 | INC R5; *enlarge #SB for H(n+1)* | 60 | DJNZ R4, WRHN; |
| 61 | RETI | | |

**OUTPUT:** $H^{(2)}, \ldots, H^{(4)} \rightarrow$ The set of hash values of subsystems

Further procedure HASHINT is given in Algorithm 8 and refers to the calculation of integral hash value $H^{(int)}$ for all subsystems of the agent. Instruction RDHN in Algorithm 8 begins reading of the $H^{(n)}$, WRHINT writes the resulting value of $H^{(int)}$.

---

**Algorithm 8:** Subroutine **HASHINT** (II step).

**INPUT:**$H^{(2)}, \ldots, H^{(4)} \leftarrow$ The set of hash functions of subsystems S$_2$, ... , S$_4$

| 1 | HASHINT: MOV R3,#255; *initial #LB* | 2 | MOV R2,#3; *number of SN* |
| 3 | RDHN: MOV R5,#89; *#field H(n)* | 4 | CLR P1.6; *enable Rg2by $\overline{CE}$* |
| 5 | SETB P1.4; *write #SB=#89 to Rg2* | 6 | CLR P1.4 |
| 7 | SETB P1.6; *lock Rg2* | 8 | MOV P2,R3; *#LB to read H(i)* |
| 9 | CLR P1.3; *enable SRAM by $\overline{CE}$* | 10 | CLR P1.1; *$\overline{OE}$ enable output H(i)* |
| 11 | MOV A,P0; *read H(i) to ACC* | 12 | SETB P1.1; *disable output of SRAM* |
| 13 | RDHN: INC R5; *go to # of next H(i+1)* | 14 | CLR P1.3; *enable SRAM by $\overline{CE}$* |
| 15 | CLR P1.1; *$\overline{OE}$ enable output H(n)* | 16 | MOV R1,P0; *read H(i) to R2* |
| 17 | SETB P1.1; *disable output of SRAM* | 18 | XOR:XRL A,R1; *Hi XOR Hi+1* |
| 19 | DEC R3; *go to next* | 20 | DJNZ R2,RDHN |
| 21 | WRHINT: MOV R5,#104; *#SB to write* | 22 | MOV P0,#R5; |
| 23 | CLR P1.6; *enable Rg2by $\overline{CE}$* | 24 | SETB P1.4; *write #104 to Rg2* |
| 25 | CLR P1.4 | 26 | SETB P1.6; *lock Rg2* |
| 27 | MOV P0,A; *output HINT from ACC* | 28 | CLR P1.3; *enable SRAM by $\overline{CE}$* |
| 29 | CLR P1.0; *enable $\overline{WR}$ HINT into SRAM* | 30 | SETB P1.0; *disable $\overline{WR}$* |
| 31 | SETB P1.3; *disable SRAM* | 32 | RETI |

**OUTPUT:** $H^{(int)} \rightarrow$ Integral hash value of subsystems S$_2$, ... , S$_4$

---

For brevity the procedure TRKEYS for distribution of fragmented keys by $S_1$ to subsystems $S_2, \ldots, S_4$ is not shown as it does not contain MVL specificity.

Imitation of subsystems $S_2, \ldots, S_4$ by the module given in Figure 6 was modelled by earlier designed module for data acquisition by ADC and transfer of obtained data array from buffer SRAM KR537RU8 to external PC. This procedure can be used for slow enough MVLL transfer to external nodes by groups of bytes. Parts of MVLL or results of ADC measurements with the length of 256 bytes can be transferred to external computer via embedded MCS-51's and interface RS232C for serial port with ADM202E chip [80]. Certainly, special conjugated C++ program was used in the receiving PC. The microassembler program for control of data acquisition and RS232C transfer module (in Figure 7) is shown in Supplement to demonstrate that given above MVLL procedures have the same level of complexity as routine ADC and RS232C software. It includes two subroutines for laboratory data acquisition from one of 4 possible sensors and forms data array in the small capacity buffer SRAM. The transfer scheme uses standard transceiver SCON and buffer register SBUF. This software illustrates quite flexible resources of MCS-51, capable e.g., to adapt 10 bits ADC to 8 bit microcontroller by forming double sequence, composed separately of senior bytes and lower bytes, containing only 2 bits. Essentially, modeling of MVL functions and MVLL schemes does not create any specific hardships for microassembler programming. The length of given above procedures is mainly determined by addressing of external SRAM by trigger registers.

However, final commutation of schemes in Figures 6 and 7 needs not only to optimize commutation of control pins, but demands modelling of decision–making module and is out of discussion here.

Microassembler subroutines for calculation of AGA operators MINIMUM, MAXIMUM, and Literal were also given earlier in [19], and MVL function calculation in C language was shown in Supplement in [18].

Brief comparative estimations of calculation time for MVL functions can be based on data given in [19], were for $k = 256$ truth levels и $n = 12$ variables the calculation needs ≈210,000 work cycles and 0.1 s at 24 MHz, where AGA operators minimally possible parameters are t $_{MIN-MAX} \approx 4$ µs and t $_{Literal} \approx 9$ µs.

Results of microassembler modeling.

- Given above 8-bit circuit boards and microassembler fragments demonstrate the principal possibility to adapt MVLL model with 21 variable and k = 256 logic levels to the small-scale network structure of 8-bit microcontrollers in the agent.
- Dual-chips scheme with external SRAM can be used as a stackable test platform for low-throughput agents for IoT and other tasks, using analog and digital sensors.
- SRAM with 1 MB capacity is quite enough to design the necessary structure of MVLL. Not less than 8 external verifiers can be used for approval of at least 256 independent entries in the MVLL. Detailed preparation of SRAM structure for MVLL provides the possibility for more quick addressing of bytes and transfer between microcontrollers. Principally, for used circuit boards SRAM can be enlarged up to 32 Mbytes.
- The distribution of fragmented keys, the acquisition of measurements data, and hashing procedures can be realized by routine microassembler programming.
- Designed methods promise new applications of QKD lines, necessary only episodically in the trusted service zone. Interaction with quantum line will strongly depend on its specification.
- In the used platform and model of MVLL the overall number of visited checkpoints in the route verification task may achieve at least 256 ones.
- Designed schemes and algorithms with 8-bit data structure can be easily commutated with further AGA modelling of blocking terms and additional data protection methods.
- It is substantial, that the structure of traditional controllers can successfully combine AGA and Boolean operators within the heterogeneous logic architecture of the agent [18].
- Detailed analysis of modern commercial wireless radiofrequency modules is needed further to choose reasonable data channels for communications with external nodes.

## 6. Discussion

The reason to propose modified version of MVLL is the desire to expand data exchange between external and internal components of the agent in order to provide more reliable data verification for prospective trust estimates and AI procedures. If the communication line of robot is not reliable or if one suspects activity of an eavesdropper, then it is quite natural to compare additionally data obtained from external storages with ones, extracted from the internal memory of the agent.

Proposed version of MVLL is represented by the rigidly given structure in SRAM, providing the design of software and the verification of correct interaction of supervisor $S_1$ with subsystems $S_2, \ldots, S_4$ in the agent. Given examples of microassembler subroutines demonstrate the possibility to realize short enough procedures by means of the small-scale hardware structure of well-known microcontrollers MCS-51. Certainly, such scheme is to be somewhat adapted to more productive controllers and FPGAs, as well as to USB controllers.

Such aspect, as interaction with modern modules for wireless communications in IoT and IoV should be investigated separately.

Another actual problem is to appreciate, if more levels (or layers) of data mixing in entries will be efficient, thus imitating more deep mixing of blocks in BC schemes.

Principally such procedure can be based just on used definition of MVLL, applied for selected variables from different entries.

Integration of protective blocking logic terms (see Section 2.4) needs to exclude logic constant $k - 1$ from active use and to reserve it for data protection service. First possible way here needs to use the shortened number of 255 truth levels instead of 256 ones, and to make necessary corrections in conjugated procedures and cycles. More original alternative here is to use heterogeneous logic architecture [18] and to apply enlarged number of truth levels in the supervisor subsystem. Here one can use e.g., 512 truth levels, described by the emulation of 16-bit calculations just in the 8-bit platform. But such method needs to choose some model of decision-making module and to design duplex mapping for subsystems with different truth levels $k$.

Separate item is the interaction and adaptation to real QKD line, and the realization of the random oracle scheme [18] with high enough capacity, adequate for practical use. Principally, in protocol [41] the random oracle is not the secret device and can serve all requests even coming from eavesdroppers!

Some discussion is necessary for the possibility to use secured link to PC during service and verification sessions.

Much more dramatic and complex discussion refers to the design of decision-making module, which is critical for the choice of SRAM structure and agents communication language, which is not specific for narrow knowledge field. From one side, loyal network nodes are to use standard network tools, but from the other side, agents in MAS should imitate people and combine universal robotic language with additional secret coding. Namely the specifics of decision making module dictates the SRAM structure and the information capacity of communication module.

Interesting aspect for further research is the possible interaction of network robotics with business management processes [81] and industrial ones [82]. Although the distant admin or the user may be far away from the robotic system, his unmanned logistics or manufacturing of goods is still in the field of view of business analytics [81], whose services also can exploit automatic systems and agents. Such interaction is served by the large number of legal norms and rules, which should be monitored and approved by network devices. That is why the task is to propose new solutions, which are suitable not only for hardware agents, but may be extended for software agents. Respectively, such methods should be compatible with common network technologies like blockchain, but besides this they should be adapted to controllers level. Further research is to show, how business technologies with customer relationship management (CRM) [81] strategy can be combined with the proposed above blockchain-induced scheme. Possible interaction with hardware robotic platforms seems to be easier to begin with a agent's customer model, taking into account such basic parameters, as customer identification and attraction. Identification here can use verification and trust estimation procedures, and attraction should be the scheme to propose adequate services for approved routes and visited objects. Certainly, such business procedures will also require secured communications, where quantum technologies can propose unique solutions.

The proposed version of MVLL potentially may be used, when the agent following the route should approve visited or prohibited for visits zones of maps, or should fix bands of acceptable parameters for some time intervals or map zones. Such models may be actual for unmanned parking systems, dumping of garbage and sludge only in specially prepared places, for loading and handling devices, for cargo security, loading monitoring systems, and for the speed control within a residential zone.

## 7. Conclusions

The extended version of MVL linked list is proposed for mobile network agents, where BC-type distributed external data storage is adapted for joint work with internal low-throughput microcontroller platforms. Modified scheme of MVLL can integrate both internal parameters of agent's subsystems and external nodes. Proposed new version

of MVLL is based on *k*-valued Allen-Givone algebra, comfort for high-dimensional data coding and verification procedures in multiparametric systems [18,19].

Parameters of internal subsystems of the mobile agent are to be approved by assigning the set of quasi-random hash values of fragmented keys, preliminary distributed by QKD line; also binary hashing *XOR* is to be applied. Traditional binary *XOR* function is chosen as the simplest hash function in microcontrollers for mixing of data of internal subsystems in the MVLL. The second step of the procedure calculates final *XOR* hash value for brevity of final data. Further obtained internal hash value is to be included into the MVLL, it may be complemented by parameters of external nodes, and is to be approved by the set of external quasi-random hash values according to earlier proposed MVLL scheme. These "fingerprints" of internal subsystems and external nodes are also can be accumulated in the memory of internal supervisor subsystem of the agent for verification in case of faults and errors, also some documented physical parameters can be kept in memory of other subsystems for check-up procedures. Such version of MVLL takes into account that free resources in internal subsystems are too limited to save large volumes of verification data.

Microassembler software examples for microcontroller platforms MCS-51 are given for basic MVLL adressing, hashing and conjugated measurements.

Specific features of route verification task are also analysed, what motivates to research further the ways to enlarge the number of logic variables in MVLL and to subdivide it into separate pages (or volumes).

The advantage of the proposed AGA based version of MVLL is that primitive logic procedures does not give the possibility to bypass any steps of data procession and to use traditional attacks; at the same time the structure of quasi-random hash values provides the unpredictability of data, necessary for verification without the disclosing of real values.

As a whole, the combination of externally approved mixed internal and external parameters is the next step to avoid trusted nodes for agents collective. Designed scheme is intended for protection of agents working distantly, when it is difficult to provide continious monitoring. MVLL storage for critical data of a robotic system is the way to approve identity of distant robots and IoT.

**Author Contributions:** Conceptualization, A.Y.B.; Methodology, A.Y.B.; Resources, N.A.V.; Software, A.Y.B.; Validation, N.A.V.; Writing—original draft, A.Y.B. All authors have read and agreed to the published version of the manuscript.

**Funding:** This research received no external funding.

**Data Availability Statement:** Not applicable.

**Conflicts of Interest:** Authors declare no conflict of interest.

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
