# Peer review of "Data Verification in the Agent, Combining Blockchain and Quantum Keys by Means of Multiple-Valued Logic"

_asi, doi:10.3390/asi6020051_

Round 1

Reviewer 1 Report

1. Author can enhanced introduction part.

2. Include  latest references. 

3. Check the grammar.

Author Response

Reply to reviewer 1

Reviewer 1. Remarks.

The authors are grateful to all reviewers for useful remarks and interesting commentaries, that will help for further work. Corresponding corrections are given further in the Table.

1. Author can enhance introduction part. 

Introduction part has included  the added sec.1.1.

1.1 Modern platforms for robotics positioning and communications

   Localization or positioning was considered in reviews [35-36]  as a process  to obtain tracked objects information concerning multiple reference points within a predetermined area, i.e. it is a procedure to identify the position of  mobile/fixed devices, including smartphones, drones, watch, beacons, and vehicles using some fixed nodes and mobile computing devices. Global navigation satellite system (GNSS) in such tasks can use signals produced by global positioning system (GPS), global navigation satellite system (GLONASS), Galileo and Beidou.  As the GPS device loses substantial power in an indoor setting due to signal attenuation, such systems cannot be used for indoor localization of devices. Alternative possibilities refer to ZigBee, Bluetooth, Radio Frequency Identification (RFID), cellular networks (including LTE and 5G), ultrawideband (UWB), frequency modulation (FM), inertial sensors and Wi-Fi. Some hybrid approaches are also possible. Most common techniques for localization include time-of-flight (TOF) measurements and received signal strength indicator (RSSI) signal measurements, utilizing the distance between known fixed stations and the target device, or fingerprint-based location estimation.

   Vehicular ad-hoc network (VANET) [37] was based on the link-layer communication (IEEE 802.11p) and has included the data exchange between the high-speed vehicles in the licensed band of 5.9 GHz (5.850–5.925 GHz). VANET differs from  other ad-hoc networks by high mobility, dynamic topology, frequent data exchange, unbounded network size, unlimited battery power, and predictable movement (which happens only on the road). It may use two types of nodes: 1) mobile nodes attached to onboard units, and 2) static nodes like traffic lamp posts, signboards, and roadside units. This network of mobile agents has provided connections vehicle-to-vehicle, vehicle-to-infrastructure, and infrastructure-to-infrastructure. Used performance metrics were hop length, minimum energy, link lifetime, route breakage, and bandwidth.

    Very actively investigated field of autonomous robotics refers to flying unmanned aerial systems (UAVs). The authors of the review [38] have analyzed flying ad-hoc networks (FANETs), which can be deployed either individually or may be incorporated into traditional wireless local area networks (WLANs).    Its main application fields include search and rescue, mailing and delivery, traffic monitoring, precision agriculture, and surveillance applications. Unmanned service of FANETs is actual in case of natural disasters, hazardous gas intrusions, wildfires, avalanches, and search of missing persons.  As routing is the most challenging job in FANETs due to such attributes of UAVs, as high mobility, 3D movement, and rapid topology changes, then a predictive method should be used for path planning and navigation in order to prevent possible collisions and to ensure the safety of the FANET. However, the data aggregated by a small UAV can be too large to be processed and stored onboard [39]. Small UAVs in FANETs also suffer from security vulnerabilities, as their limited storage and computing capabilities do not allow to perform computational-intensive tasks locally [40,41]. An intruder intending to attack the network UAV [38] can transmit massive reservation requests, eavesdrop  instructions, and modify the information. UAVs connected to Wi-Fi are considered as less secured in comparison with cellular networks, due to unreliable links and poor security methods. False transmitter can be attached to a UAV and may send fake instructions, in addition to this UAVs can become a luring target for physical attacks [39,41]. In such instances, an attacker can dissemble the captured UAV to get access to internal data via interfaces and USB ports.

     GPS spoofing, see e.g., [38] is another major security threat for small UAVs. An adversary can transmit fake GPS signals to an intended UAV with enlarged power than the actual GPS signals. Thus, localization system must verify actual positions of neighboring UAVs and associated distances in order to avoid the GPS spoofing attacks.

     The detection and identification of vulnerabilities for UAVs refers to popular short-range wireless networking technologies like Wi-Fi (IEEE 802.11), ZigBee (IEEE 802.15.4), Bluetooth (IEEE 802.15.1),  LoRaWAN, and Sigfox [38,42], differing by the range and the throughput for licensed or unlicensed spectrum types. Wi-Fi provides a set of specifications for radio bands of 2.4, 3.6, 5, and 60 GHz. IEEE 802.11a/b/g/n/ac is the first choice to provide the transmission of medium size video and images for distances of approximately 100 m, but unlicensed versions can provide up to few hundred meters. A multi-hop networking scheme may expand the transmission range to kilometers. An alternative to Wi-Fi is the use of low-cost and low-power methods like Bluetooth and ZigBee. Bluetooth (IEE 802.15.1) is a low cost and low power variant, which operates in an unlicensed band of 2.4 GHz with a contact range of 10 to 100 m and uses a distributed frequency-hopping transmission spectrum.

     Licensed 5G and 6G generation technologies are expected  [38] to offer improved data rates and coverages in linking of FANETs, to provide high device mobility and integration of a massive number of UAVs in an ultra-reliable way, to serve multi-access edge computing, and to incorporate cloud computing. Low-power wide area networks (LPWAN) can be another good option for UAVs which consumes less energy and offers a wide range of connectivity. LPWAN allows transmitting data for a longer duration of time and without much loss of energy resources. For IoT users, LoRaWAN [43] has been designed as a technology for the management of low energy consumption transmissions, using a novel network paradigm for bidirectional connectivity, localization, and mobility management. It provides a new framework for LPWAN execution, providing long-range communications in the band 868/900 MHz with data rates ranging from 0.3 kbps to 50 kbps and network coverage from 5 to 15 km.  Sigfox, similar to LoRaWAN, is a low-speed, but low-power and long-range solution for UAVs, it supports open-sight up to 30 km of range in the same band as LoRaWAN.

    Another modern trend for enhancement of data privacy  and integrity in UAV communication networks is the aerial blockchain, especially supported by 5G and 6G [38,44]. Blockchain-based software for UAV is expected to provide flexibility, dynamics, and on-the-fly decision capabilities. UAVs  can be integrated potentially into the Internet network, providing access to cloud computing and web technologies for the realization of smart IoT systems.

2. Include 

   latest references. 

2. A dozen of recent references 2022-2023  were added; some of them were given in the new sec.1, the other ones has replaced more “old” papers and editions of books.

3. Check the grammar.

3.Text corrections were done, yellow marks  correction are in the revised version.

Reviewer 2 Report

Thank you very much for the opportunity to review this paper. In general, I see it as a very good candidate to be published in this Journal.

The Introduction is clear and extensive, I have nothing to object to. A very good reference structure is used for the aim of the paper and the bibliographical references that support it (classified into subsections). The literature review is therefore also complete and well defined. I would only recommend seeing there a reference to the relevance of looking at systems like customer relationship management (CRM) as one of the most powerful tools to evaluate the desires, expectations, and needs of entities that are supposed to use these processes and derivatives, which I consider extremely relevant and miss in this study. There are several examples of the use of these information systems for non-commercial purposes but technological innovative purposes (I especially recommend that you please review and make citation of https://doi.org/10.1007/s11365-022-00800-x).

The methodology and the Empirical Results are shown in this paper with clarity of exposition and with great care in their justification and relationship between them, for what I consider to be an exceptionally exposed work. I have found no inconsistencies or errors (I admit I am not a specialist on this subject, but despite all this I have been able to follow the course of the exhibition without problems). The graphs and tables greatly help to understand and follow the research process that has been carried out.

The discussion and conclusions sections are clear and relevant, well founded on the results.

References are sufficient and relevant, although I would recommend the acceptance of this paper after including the reference and details of CRM review recommended literature.

The supplementagry materials are of great value.

In summary, I recommend the acceptance of the paper after minor revision.

Author Response

Reply to reviewer 2

Reviewer 2. Remarks.

    The authors are grateful to all reviewers for useful remarks and interesting commentaries, that will help for further work. Corresponding corrections are shown further in the Table. The idea to combine CRM for robotic services seems  “fresh” and fruitful for further! Some possible steps are commented in sec 5. Discussion.

I would only recommend seeing there a reference to the relevance of looking at systems like customer relationship management (CRM) as one of the most powerful tools … which I consider extremely relevant and miss in this study. …I especially recommend …https://doi.org/10.1007/s11365-022-00800-x).

  “Interesting aspect for further research is the possible interaction of network robotics with business management processes [81] and industrial ones [82]. Although the distant admin or the user may be far away from the robotic system, his unmanned logistics or manufacturing of goods is still in the field of view of business analytics [81], whose services also can exploit automatic systems and agents. Such interaction is served by the  large number of legal norms and rules, which should be monitored and approved by network devices. That is why the task is to propose new solutions, which are suitable not only for hardware agents, but may be extended for software agents. Respectively, such methods should be compatible with common  network technologies  like blockchain, but besides this they should be adapted to controllers level. Further research is to show, how business  technologies  with customer relationship management (CRM) [81] strategy  can be combined with  the proposed above blockchain-induced scheme. Possible interaction with hardware robotic platforms seems to be easier to begin with a  agent`s customer model, taking into account such basic parameters,  as  customer identification and  attraction. Identification here can use verification and trust estimation procedures, and attraction should be the scheme to propose adequate services for approved routes and visited objects. Certainly, such business procedures will also require secured communications, where quantum technologies can propose unique solutions. “

Reviewer 3 Report

please split figure 2 in to two figures since it is very condensed

also please redraw figure 5 clearly

if you please give small tutorial about MVL after the references

Author Response

Reply to reviewer 3

Reviewer 3. Remarks.

    The authors are grateful to all reviewers for useful remarks and interesting commentaries, that will help for further work. Corresponding corrections are shown further in the Table.

1. Please split figure 2 in to two figures since it is very condensed

1. Fig. 2 was splitted  into figs. 2 and 3 in the revised manuscript.

2.Please redraw figure 5 clearly.

 2. Fig 5 (in revision it is  fig. 6) was modified, we hope that this will make it more comfort for readers.

3. Please give small tutorial about MVL after the references

3. The brief tutorial was added as section S1 to complementary materials.

S1. Brief tutorial to multiple-valued logic

   This brief commentary was added for readers, who hasn`t access to  MVL profile library resources.

   The detailed and clear enough introductions and reviews were recently given in Open Access papers in Quantum reports (mdpi) [18] and Applied Sciences (mdpi) [19].   The original resource here is the second edition of the book [50], which contains several chapters disclosing the history and concepts of MVL.

     Allen-Givone algebra is used in the given paper as one of possible MVL models. Principally, the methods to use discrete system with many levels of truth substantially resembles the methods of the two-level Boolean logic with 0 s and 1s, as it also uses truth tables, product terms (often called minterms in the Boolean one), and minimization procedures. Several complete sets including AGA are known, which guarantee the representation of arbitrarily given MVL function as a some combination of constants, Min, Max and Literals X(a,b). That property provides the possibility to represent correctly any complicated control process by such operators, what is very substantial for data processing schemes. But the drawback is that such expressions may be bulky and the calculation time may be large and not always exactly predictable beforehand, if one doesn`t know the specifics of data. However, modern computers has the enhanced throughput and can carry out such calculations for many useful tasks.

     MVL can describe and navigate systems having much greater number of states than Boolean logic, what can be directly seen from the MVL truth table, but on practice for debugging of procedures one can also directly modify parameters (a,b) in Literals X(a,b), written in logic expressions. That is why one doesn`t need to re-write the truth table every time, but such possibility needs accuracy and some experience. That is why modern large-scale and global systems can be compact described by MVL logic expressions, what makes it the attractive tool. However, as the “free cheese can lead to a mousetrap”, the pay here is the wasteful minimization procedure, which leads to multi-criteria optimization task. Respectively, the design of complicated MVL model for agents is the difficult task.

     MVL gives interesting possibilities to model data protection systems and secret coding schemes, which were presented in the review [12]. Such methods need good sources of random numbers, what naturally leads to quantum key distribution and quantum random number generations. For example, the method to describe quantum protocol [51] with entangled photon pairs by AGA logic expressions was proposed in [19]; it is intended for route-verification of a mobile robot.

     Min/Max operators are very close to ones used in the more specific fuzzy logic, that is why it is easier to combine namely them. In many papers and books, the fuzzy logic is being called the multi-level logic, although the fuzzy logic by L.Zadeh uses infinite description of the truth, but any computer emulation of the  fuzzy logic is certainly the discrete one. That creates some confusion of terms; that is why the authors prefer to accent discrete MVL calculus for AGA. In any way, fuzzy logic is a very popular technology for control systems, what was one of the motives to propose heterogeneous architecture of the intellectual agent [18], intended for MVL control schemes in systems, combining fuzzy controllers with their approximate and “ quick” models  with other regulators. 

    One should also take into account, that many traditional discrete MVL models with three truth levels were investigated, see  e.g., referencies to [53-54]. However, the authors follow the opinion, that discrete systems with large numbers of truth-levels (256 and more) are much more interesting for the applied sphere, and C. Allen and D. Givone not for nothing has named their logic model in [50] namely the application–oriented one.
